# Early-Learning Regularization Prevents Memorization of Noisy Labels

**Sheng Liu**
Center for Data Science
New York University
shengliu@nyu.edu

**Jonathan Niles-Weed**
Center for Data Science, and
Courant Inst. of Mathematical Sciences
New York University
jnw@cims.nyu.edu

**Narges Razavian**
Department of Population Health, and
Department of Radiology
NYU School of Medicine
narges.razavian@nyulangone.org

**Carlos Fernandez-Granda**
Center for Data Science, and
Courant Inst. of Mathematical Sciences
New York University
cfgranda@cims.nyu.edu

## Abstract

We propose a novel framework to perform classification via deep learning in the presence of noisy annotations. When trained on noisy labels, deep neural networks have been observed to first fit the training data with clean labels during an "early learning" phase, before eventually memorizing the examples with false labels. We prove that early learning and memorization are fundamental phenomena in high-dimensional classification tasks, even in simple linear models, and give a theoretical explanation in this setting. Motivated by these findings, we develop a new technique for noisy classification tasks, which exploits the progress of the early learning phase. In contrast with existing approaches, which use the model output during early learning to detect the examples with clean labels, and either ignore or attempt to correct the false labels, we take a different route and instead capitalize on early learning via regularization. There are two key elements to our approach. First, we leverage semi-supervised learning techniques to produce target probabilities based on the model outputs. Second, we design a regularization term that steers the model towards these targets, implicitly preventing memorization of the false labels. The resulting framework is shown to provide robustness to noisy annotations on several standard benchmarks and real-world datasets, where it achieves results comparable to the state of the art.

## 1 Introduction

Deep neural networks have become an essential tool for classification tasks [19, 15, 11]. These models tend to be trained on large curated datasets such as CIFAR-10 [18] or ImageNet [9], where the vast majority of labels have been manually verified. Unfortunately, in many applications such datasets are not available, due to the cost or difficulty of manual labeling (e.g. [13, 32, 25, 1]). However, datasets with lower quality annotations, obtained for instance from online queries [5] or crowdsourcing [49, 53], may be available. Such annotations inevitably contain numerous mistakes or *label noise*. It is therefore of great importance to develop methodology that is robust to the presence of noisy annotations.

When trained on noisy labels, deep neural networks have been observed to first fit the training data with clean labels during an *early learning* phase, before eventually *memorizing* the examples with

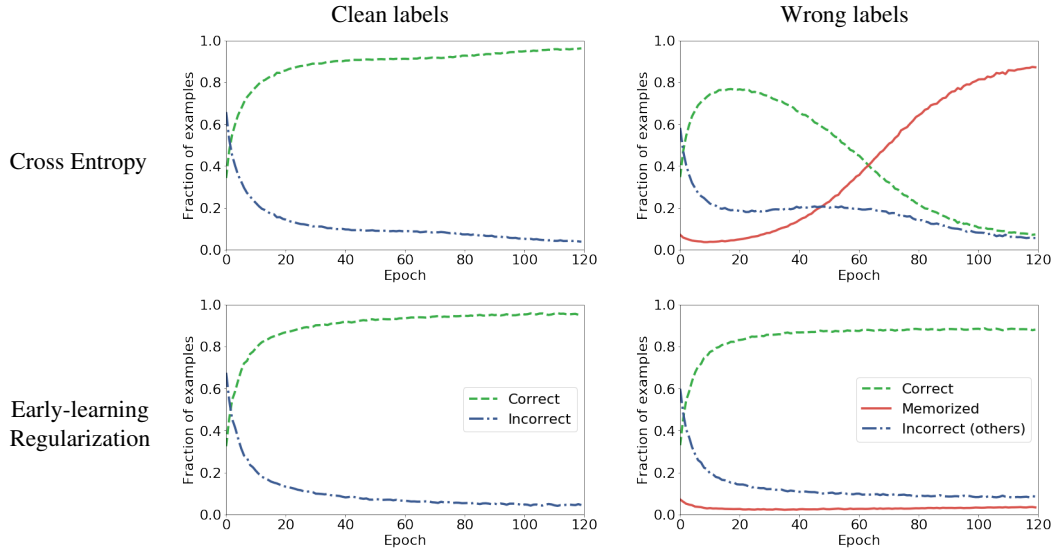

Figure 1: Results of training a ResNet-34 [15] neural network with a traditional cross entropy loss (top row) and our proposed method (bottom row) to perform classification on the CIFAR-10 dataset where 40% of the labels are flipped at random. The left column shows the fraction of examples with clean labels that are predicted correctly (green) and incorrectly (blue). The right column shows the fraction of examples with wrong labels that are predicted correctly (green), *memorized* (the prediction equals the wrong label, shown in red), and incorrectly predicted as neither the true nor the labeled class (blue). The model trained with cross entropy begins by learning to predict the true labels, even for many of the examples with wrong label, but eventually memorizes the wrong labels. Our proposed method based on early-learning regularization prevents memorization, allowing the model to continue learning on the examples with clean labels to attain high accuracy on examples with both clean and wrong labels.

false labels [3, 54]. In this work we study this phenomenon and introduce a novel framework that exploits it to achieve robustness to noisy labels. Our main contributions are the following:

- In Section 3 we establish that early learning and memorization are fundamental phenomena in high dimensions, proving that they occur even for simple linear generative models.

- In Section 4 we propose a technique that utilizes the early-learning phenomenon to counteract the influence of the noisy labels on the gradient of the cross entropy loss. This is achieved through a regularization term that incorporates target probabilities estimated from the model outputs using several semi-supervised learning techniques.

- In Section 6 we show that the proposed methodology achieves results comparable to the state of the art on several standard benchmarks and real-world datasets. We also perform a systematic ablation study to evaluate the different alternatives to compute the target probabilities, and the effect of incorporating mixup data augmentation [55].

## 2   Related Work

In this section we describe existing techniques to train deep-learning classification models using data with noisy annotations. We focus our discussion on methods that do not assume the availability of small subsets of training data with clean labels (as opposed, for example, to [16, 34, 41]). We also assume that the correct classes are known (as opposed to [44]).

*Robust-loss* methods propose cost functions specifically designed to be robust in the presence of noisy labels. These include Mean Absolute Error (MAE) [10], Improved MAE [43], which is a reweighted MAE, Generalized Cross Entropy [56], which can be interpreted as a generalization of MAE, Symmetric Cross Entropy [45], which adds a reverse cross-entropy term to the usual cross-entropy loss, and $\mathcal{L}_{\text{DIM}}$ [48], which is based on information-theoretic considerations. *Loss-correction*

methods explicitly correct the loss function to take into account the noise distribution, represented by a transition matrix of mislabeling probabilities [31, 12, 46, 39].

Robust-loss and loss-correction techniques do not exploit the early-learning phenomenon mentioned in the introduction. This phenomenon was described in [3] (see also [54]), and analyzed theoretically in [23]. Our theoretical approach differs from theirs in two respects. First, Ref. [23] focus on a least squares regression task, whereas we focus on the noisy label problem in classification. Second, and more importantly, we prove that early learning and memorization occur even in a *linear* model.

Early learning can be exploited through *sample selection*, where the model output during the early-learning stage is used to predict which examples are mislabeled and which have been labeled correctly. The prediction is based on the observation that mislabeled examples tend to have higher loss values. Co-teaching [14, 52] performs sample selection by using two networks, each trained on a subset of examples that have a small training loss for the other network (see [17, 28] for related approaches). A limitation of this approach is that the examples that are selected tend to be *easier*, in the sense that the model output during early learning approaches the true label. As a result, the gradient of the cross-entropy with respect to these examples is small, which slows down learning [6]. In addition, the subset of selected examples may not be rich enough to generalize effectively to held-out data [35].

An alternative to sample selection is *label correction*. During the early-learning stage the model predictions are accurate on a subset of the mislabeled examples (see the top row of Figure 1). This suggests correcting the corresponding labels. This can be achieved by computing new labels equal to the probabilities estimated by the model (known as *soft labels*) or to one-hot vectors representing the model predictions (*hard labels*) [38, 51]. Another option is to set the new labels to equal a convex combination of the noisy labels and the soft or hard labels [33]. Label correction is usually combined with some form of iterative sample selection [2, 27, 35, 22] or with additional regularization terms [38]. SELFIE [35] uses label replacement to correct a subset of the labels selected by considering past model outputs. Ref. [27] computes a different convex combination with hard labels for each example based on a measure of model dimensionality. Ref. [2] fits a two-component mixture model to carry out sample selection, and then corrects labels via convex combination as in [33]. They also apply mixup data augmentation [55] to enhance performance. In a similar spirit, DivideMix [22] uses two networks to perform sample selection via a two-component mixture model, and applies the semi-supervised learning technique MixMatch [4].

Our proposed approach is somewhat related in spirit to label correction. We compute a probability estimate that is analogous to the soft labels mentioned above, and then exploit it to avoid memorization. However it is also fundamentally different: instead of modifying the labels, we propose a novel regularization term explicitly designed to correct the gradient of the cross-entropy cost function. This yields strong empirical performance, without needing to incorporate sample selection.

## 3 Early learning as a general phenomenon of high-dimensional classification

As the top row of Figure 1 makes clear, deep neural networks trained with noisy labels make progress during the early learning stage before memorization occurs. In this section, we show that far from being a peculiar feature of deep neural networks, this phenomenon is intrinsic to high-dimensional classification tasks, even in the simplest setting. Our theoretical analysis is also the inspiration for the early-learning regularization procedure we propose in Section 4.

We exhibit a simple *linear* model with noisy labels which evinces the same behavior as described above: the *early learning* stage, when the classifier learns to correctly predict the true labels, even on noisy examples, and the *memorization* stage, when the classifier begins to make incorrect predictions because it memorizes the wrong labels. This is illustrated in Figure A.1, which demonstrates that empirically the linear model has the same qualitative behavior as the deep-learning model in Figure 1. We show that this behavior arises because, early in training, the gradients corresponding to the correctly labeled examples dominate the dynamics—leading to early progress towards the true optimum—but that the gradients corresponding to wrong labels soon become dominant—at which point the classifier simply learns to fit the noisy labels.

We consider data drawn from a mixture of two Gaussians in $\mathbb{R}^p$. The (clean) dataset consists of $n$ i.i.d. copies of $(\mathbf{x}, \mathbf{y}^*)$. The label $\mathbf{y}^* \in \{0, 1\}^2$ is a one-hot vector representing the cluster assignment,

and

$$\mathbf{x} \sim \mathcal{N}(+\mathbf{v}, \sigma^2 I_{p \times p}) \quad \text{if } \mathbf{y}^* = (1, 0)$$
$$\mathbf{x} \sim \mathcal{N}(-\mathbf{v}, \sigma^2 I_{p \times p}) \quad \text{if } \mathbf{y}^* = (0, 1),$$

where $\mathbf{v}$ is an arbitrary unit vector in $\mathbb{R}^p$ and $\sigma^2$ is a small constant. The optimal separator between the two classes is a hyperplane through the origin perpendicular to $\mathbf{v}$. We focus on the setting where $\sigma^2$ is fixed while $n, p \to \infty$. In this regime, the classification task is nontrivial, since the clusters are, approximately, two spheres whose centers are separated by 2 units with radii $\sigma\sqrt{p} \gg 2$.

We only observe a dataset with noisy labels $(\mathbf{y}^{[1]}, \ldots, \mathbf{y}^{[n]})$,

$$\mathbf{y}^{[i]} = \begin{cases} (\mathbf{y}^*)^{[i]} & \text{with probability } 1 - \Delta \\ \tilde{\mathbf{y}}^{[i]} & \text{with probability } \Delta, \end{cases} \tag{1}$$

where $\{\tilde{\mathbf{y}}^{[i]}\}_{i=1}^n$ are i.i.d. random one-hot vectors which take values $(1, 0)$ and $(0, 1)$ with equal probability.

We train a linear classifier by gradient descent on the cross entropy:

$$\min_{\Theta \in \mathbb{R}^{2 \times p}} \mathcal{L}_{\text{CE}}(\Theta) := -\frac{1}{n} \sum_{i=1}^n \sum_{c=1}^2 \mathbf{y}_c^{[i]} \log(\mathcal{S}(\Theta \mathbf{x}^{[i]})_c) \,,$$

where $\mathcal{S} : \mathbb{R}^2 \to [0, 1]^2$ is a softmax function. In order to separate the true classes well (and not overfit to the noisy labels), the rows of $\Theta$ should be correlated with the vector $\mathbf{v}$.

The gradient of this loss with respect to the model parameters $\Theta$ corresponding to class $c$ reads

$$\nabla \mathcal{L}_{\text{CE}}(\Theta)_c = \frac{1}{n} \sum_{i=1}^n \mathbf{x}^{[i]} \left( \mathcal{S}(\Theta \mathbf{x}^{[i]})_c - \mathbf{y}_c^{[i]} \right), \tag{2}$$

Each term in the gradient therefore corresponds to a weighted sum of the examples $\mathbf{x}^{[i]}$, where the weighting depends on the agreement between $\mathcal{S}(\Theta \mathbf{x}^{[i]})_c$ and $\mathbf{y}_c^{[i]}$.

Our main theoretical result shows that this linear model possesses the properties described above. During the early-learning stage, the algorithm makes progress and the accuracy on wrongly labeled examples increases. However, during this initial stage, the relative importance of the wrongly labeled examples continues to grow; once the effect of the wrongly labeled examples begins to dominate, memorization occurs.

**Theorem 1** (Informal). *Denote by $\{\Theta_t\}$ the iterates of gradient descent with step size $\eta$. For any $\Delta \in (0, 1)$, there exists a constant $\sigma_\Delta$ such that, if $\sigma \leq \sigma_\Delta$ and $p/n \in (1 - \Delta/2, 1)$, then with probability $1 - o(1)$ as $n, p \to \infty$ there exists a $T = \Omega(1/\eta)$ such that:*

- *Early learning succeeds: For $t < T$, $-\nabla \mathcal{L}(\Theta_t)$ is well correlated with the correct separator $\mathbf{v}$, and at $t = T$ the classifier has higher accuracy on the wrongly labeled examples than at initialization.*

- *Gradients from correct examples vanish: Between $t = 0$ and $t = T$, the magnitudes of the coefficients $\left( \mathcal{S}(\Theta_t \mathbf{x}^{[i]})_c - \mathbf{y}_c^{[i]} \right)$ corresponding to examples with clean labels decreases while the magnitudes of the coefficients for examples with wrong labels increases.*

- *Memorization occurs: As $t \to \infty$, the classifier $\Theta_t$ memorizes all noisy labels.*

Due to space constraints, we defer the formal statement of Theorem 1 and its proof to the supplementary material.

The proof of Theorem 1 is based on two observations. First, while $\Theta$ is still not well correlated with $\mathbf{v}$, the coefficients $\mathcal{S}(\Theta \mathbf{x}^{[i]})_c - \mathbf{y}_c^{[i]}$ are similar for all $i$, so that $\nabla \mathcal{L}_{\text{CE}}$ points approximately in the average direction of the examples. Since the majority of data points are correctly labeled, this means the gradient is still well correlated with the correct direction during the early learning stage. Second, once $\Theta$ becomes correlated with $\mathbf{v}$, the gradient begins to point in directions orthogonal to the correct

direction $\mathbf{v}$; when the dimension is sufficiently large, there are enough of these orthogonal directions to allow the classifier to completely memorize the noisy labels.

This analysis suggests that in order to learn on the correct labels and avoid memorization it is necessary to (1) ensure that the contribution to the gradient from examples with clean labels remains large, and (2) neutralize the influence of the examples with wrong labels on the gradient. In Section 4 we propose a method designed to achieve this via regularization.

## 4 Methodology

### 4.1 Gradient analysis of softmax classification from noisy labels

In this section we explain the connection between the linear model from Section 3 and deep neural networks. Recall the gradient of the cross-entropy loss with respect to $\Theta$ given in (2). Performing gradient descent modifies the parameters iteratively to push $\mathcal{S}(\Theta\mathbf{x}^{[i]})$ closer to $\mathbf{y}^{[i]}$. If $c$ is the true class so that $\mathbf{y}_c^{[i]} = 1$, the contribution of the $i$th example to $\nabla\mathcal{L}_{\mathrm{CE}}(\Theta)_c$ is aligned with $-\mathbf{x}^{[i]}$, and gradient descent moves in the direction of $\mathbf{x}^{[i]}$. However, if the label is noisy and $\mathbf{y}_c^{[i]} = 0$, then gradient descent moves in the opposite direction, which eventually leads to memorization as established by Theorem 1.

We now show that for nonlinear models based on neural networks, the effect of label noise is analogous. We consider a classification problem with $C$ classes, where the training set consists of $n$ examples $\{\mathbf{x}^{[i]}, \mathbf{y}^{[i]}\}_{i=1}^n$, $\mathbf{x}^{[i]} \in \mathbb{R}^d$ is the $i$th input and $\mathbf{y}^{[i]} \in \{0,1\}^C$ is a one-hot label vector indicating the corresponding class. The classification model maps each input $\mathbf{x}^{[i]}$ to a $C$-dimensional encoding using a deep neural network $\mathcal{N}_{\mathbf{x}^{[i]}}(\Theta)$ and then feeds the encoding into a softmax function $\mathcal{S}$ to produce an estimate $\mathbf{p}^{[i]}$ of the conditional probability of each class given $\mathbf{x}^{[i]}$,

$$\mathbf{p}^{[i]} := \mathcal{S}\left(\mathcal{N}_{\mathbf{x}^{[i]}}(\Theta)\right). \tag{3}$$

$\Theta$ denotes the parameters of the neural network. The gradient of the cross-entropy loss,

$$\mathcal{L}_{\mathrm{CE}}(\Theta) := -\frac{1}{n}\sum_{i=1}^n\sum_{c=1}^C \mathbf{y}_c^{[i]}\log\mathbf{p}_c^{[i]}, \tag{4}$$

with respect to $\Theta$ equals

$$\nabla\mathcal{L}_{\mathrm{CE}}(\Theta) = \frac{1}{n}\sum_{i=1}^n \nabla\mathcal{N}_{\mathbf{x}^{[i]}}(\Theta)\left(\mathbf{p}^{[i]} - \mathbf{y}^{[i]}\right), \tag{5}$$

where $\nabla\mathcal{N}_{\mathbf{x}^{[i]}}(\Theta)$ is the Jacobian matrix of the neural-network encoding for the $i$th input with respect to $\Theta$. Here we see that label noise has the same effect as in the simple linear model. If $c$ is the true class, but $\mathbf{y}_c^{[i]} = 0$ due to the noise, then the contribution of the $i$th example to $\nabla\mathcal{L}_{\mathrm{CE}}(\Theta)_c$ is reversed. The entry corresponding to the *impostor* class $c'$, is also reversed because $\mathbf{y}_{c'}^{[i]} = 1$. As a result, performing stochastic gradient descent eventually results in memorization, as in the linear model (see Figures 1 and A.1). Crucially, the influence of the label noise on the gradient of the cross-entropy loss is restricted to the term $\mathbf{p}^{[i]} - \mathbf{y}^{[i]}$ (see Figure B.1). In Section 4.2 we describe how to counteract this influence by exploiting the early-learning phenomenon.

### 4.2 Early-learning regularization

In this section we present a novel framework for learning from noisy labels called early-learning regularization (ELR). We assume that we have available a *target*[1] vector of probabilities $\mathbf{t}^{[i]}$ for each example $i$, which is computed using past outputs of the model. Section 4.3 describes several techniques to compute the targets. Here we explain how to use them to avoid memorization.

Due to the early-learning phenomenon, we assume that at the beginning of the optimization process the targets do not overfit the noisy labels. ELR exploits this using a regularization term that seeks to

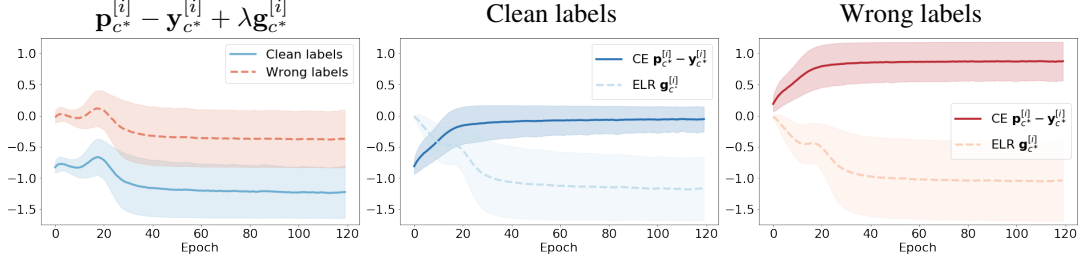

Figure 2: Illustration of the effect of the regularization on the gradient of the ELR loss (see Lemma 2) for the same deep-learning model as in Figure 1. On the left, we plot the entry of $\mathbf{p}^{[i]} - \mathbf{y}^{[i]} + \lambda\mathbf{g}^{[i]}$ corresponding to the true class, denoted by $c^*$, for training examples with clean (blue) and wrong (red) labels. The center image shows the $c^*$th entry of the cross-entropy (CE) term $\mathbf{p}^{[i]} - \mathbf{y}^{[i]}$ (dark blue) and the regularization term $\mathbf{g}^{[i]}$ (light blue) separately for the examples with clean labels. During early learning the CE term dominates, but afterwards it vanishes as the model learns the clean labels (i.e. $\mathbf{p}^{[i]} \approx \mathbf{y}^{[i]}$). However, the regularization term compensates for this, forcing the model to continue learning mainly on the examples with clean labels. On the right, we show the CE and the regularization term (dark and light red respectively) separately for the examples with wrong labels. The regularization cancels out the CE term, preventing memorization. In all plots the curves represent the mean value, and the shaded regions are within one standard deviation of the mean.

maximize the inner product between the model output and the targets,

$$\mathcal{L}_{\text{ELR}}(\Theta) := \mathcal{L}_{\text{CE}}(\Theta) + \frac{\lambda}{n}\sum_{i=1}^{n}\log\left(1 - \langle\mathbf{p}^{[i]}, \mathbf{t}^{[i]}\rangle\right). \tag{6}$$

The logarithm in the regularization term counteracts the exponential function implicit in the softmax function in $\mathbf{p}^{[i]}$. A possible alternative to this approach would be to penalize the Kullback-Leibler divergence between the model outputs and the targets. However, this does not exploit the early-learning phenomenon effectively, because it leads to overfitting the targets as demonstrated in Section C.

The key to understanding why ELR is effective lies in its gradient, derived in the following lemma, which is proved in Section E.

**Lemma 2** (Gradient of the ELR loss). *The gradient of the loss defined in Eq. (6) is equal to*

$$\nabla\mathcal{L}_{ELR}(\Theta) = \frac{1}{n}\sum_{i=1}^{n}\nabla\mathcal{N}_{\mathbf{x}^{[i]}}(\Theta)\left(\mathbf{p}^{[i]} - \mathbf{y}^{[i]} + \lambda\mathbf{g}^{[i]}\right) \tag{7}$$

*where the entries of $\mathbf{g}^{[i]} \in \mathbb{R}^C$ are given by*

$$\mathbf{g}_c^{[i]} := \frac{\mathbf{p}_c^{[i]}}{1 - \langle\mathbf{p}^{[i]}, \mathbf{t}^{[i]}\rangle}\sum_{k=1}^{C}(\mathbf{t}_k^{[i]} - \mathbf{t}_c^{[i]})\mathbf{p}_k^{[i]}, \qquad 1 \le c \le C. \tag{8}$$

In words, the sign of $\mathbf{g}_c^{[i]}$ is determined by a weighted combination of the difference between $\mathbf{t}_c^{[i]}$ and the rest of the entries in the target.

If $c^*$ is the true class, then the $c^*$th entry of $\mathbf{t}^{[i]}$ tends to be dominant during early-learning. In that case, the $c^*$th entry of $\mathbf{g}^{[i]}$ is negative. This is useful both for examples with clean labels and for those with wrong labels. For examples with clean labels, the cross-entropy term $\mathbf{p}^{[i]} - \mathbf{y}^{[i]}$ tends to vanish after the early-learning stage because $\mathbf{p}^{[i]}$ is very close to $\mathbf{y}^{[i]}$, allowing examples with wrong labels to dominate the gradient. Adding $\mathbf{g}^{[i]}$ counteracts this effect by ensuring that the magnitudes of the coefficients on examples with clean labels remains large. The center image of Figure 2 shows this effect. For examples with wrong labels, the cross entropy term $\mathbf{p}_{c^*}^{[i]} - \mathbf{y}_{c^*}^{[i]}$ is positive because $\mathbf{y}_{c^*}^{[i]} = 0$. Adding the negative term $\mathbf{g}_{c^*}^{[i]}$ therefore dampens the coefficients on these mislabeled examples, thereby diminishing their effect on the gradient (see right image in Figure 2). Thus, ELR fulfils the two desired properties outlined at the end of Section 3: boosting the gradient of examples with clean labels, and neutralizing the gradient of the examples with false labels.

| Datasets (Architecture) | Methods | Symmetric label noise | | | | Asymmetric label noise | | | |
|---|---|---|---|---|---|---|---|---|---|
| | | 20% | 40% | 60% | 80% | 10% | 20% | 30% | 40% |
| CIFAR10 (ResNet34) | Cross entropy | 86.98 ± 0.12 | 81.88 ± 0.29 | 74.14 ± 0.56 | 53.82 ± 1.04 | 90.69 ± 0.17 | 88.59 ± 0.34 | 86.14 ± 0.40 | 80.11 ± 1.44 |
| | Bootstrap [33] | 86.23 ± 0.23 | 82.23 ± 0.37 | 75.12 ± 0.56 | 54.12 ± 1.32 | 90.32 ± 0.21 | 88.26 ± 0.24 | 86.57 ± 0.35 | 81.21 ± 1.47 |
| | Forward [31] | 87.99 ± 0.36 | 83.25 ± 0.38 | 74.96 ± 0.65 | 54.64 ± 0.44 | 90.52 ± 0.26 | 89.09 ± 0.47 | 86.79 ± 0.36 | 83.55 ± 0.58 |
| | GSE [56] | 89.83 ± 0.20 | 87.13 ± 0.22 | 82.54 ± 0.23 | 64.07 ± 1.38 | 90.91 ± 0.22 | 89.33 ± 0.17 | 85.45 ± 0.74 | 76.74 ± 0.61 |
| | SL [45] | 89.83 ± 0.32 | 87.13 ± 0.26 | 82.81 ± 0.61 | 68.12 ± 0.81 | 91.72 ± 0.31 | 90.44 ± 0.27 | 88.48 ± 0.46 | 82.51 ± 0.45 |
| | ELR | **91.16 ± 0.08** | **89.15 ± 0.17** | **86.12 ± 0.49** | **73.86 ± 0.61** | **93.27 ± 0.11** | **93.52 ± 0.23** | **91.89 ± 0.22** | **90.12 ± 0.47** |
| | ELR* | **92.12 ± 0.35** | **91.43 ± 0.21** | **88.87 ± 0.24** | **80.69 ± 0.57** | **94.57 ± 0.23** | **93.28 ± 0.19** | **92.70 ± 0.41** | **90.35 ± 0.38** |
| CIFAR100 (ResNet34) | Cross entropy | 58.72 ± 0.26 | 48.20 ± 0.65 | 37.41 ± 0.94 | 18.10 ± 0.82 | 66.54 ± 0.42 | 59.20 ± 0.18 | 51.40 ± 0.16 | 42.74 ± 0.61 |
| | Bootstrap [33] | 58.27 ± 0.21 | 47.66 ± 0.55 | 34.68 ± 1.1 | 21.64 ± 0.97 | 67.27 ± 0.78 | 62.14 ± 0.32 | 52.87 ± 0.19 | 45.12 ± 0.57 |
| | Forward [31] | 39.19 ± 2.61 | 31.05 ± 1.44 | 19.12 ± 1.95 | 8.99 ± 0.58 | 45.96 ± 1.21 | 42.46 ± 2.16 | 38.13 ± 2.97 | 34.44 ± 1.93 |
| | GSE [56] | 66.81 ± 0.42 | 61.77 ± 0.24 | 53.16 ± 0.78 | 29.16 ± 0.74 | 68.36 ± 0.42 | 66.59 ± 0.22 | 61.45 ± 0.26 | 47.22 ± 1.15 |
| | SL [45] | 70.38 ± 0.13 | 62.27 ± 0.22 | 54.82 ± 0.57 | 25.91 ± 0.44 | 73.12 ± 0.22 | 72.56 ± 0.22 | 72.12 ± 0.24 | 69.32 ± 0.87 |
| | ELR | **74.21 ± 0.22** | **68.28 ± 0.31** | **59.28 ± 0.67** | **29.78 ± 0.56** | **74.20 ± 0.31** | **74.03 ± 0.31** | **73.71 ± 0.22** | **73.26 ± 0.64** |
| | ELR* | **74.68 ± 0.31** | **68.43 ± 0.42** | **60.05 ± 0.78** | **30.27 ± 0.86** | **74.52 ± 0.32** | **74.20 ± 0.25** | **74.02 ± 0.33** | **73.73 ± 0.34** |

⋆ *Results with cosine annealing learning rate.*

Table 1: Comparison with state-of-the-art methods on CIFAR-10 and CIFAR-100 with symmetric and asymmetric label noise. The bootstrap and SL methods were reimplemented using publicly available code, the rest of results are taken from [56]. The mean accuracy and its standard deviation are computed over five noise realizations.

## 4.3 Target estimation

ELR requires a target probability for each example in the training set. The target can be set equal to the model output, but using a running average is more effective. In semi-supervised learning, this technique is known as temporal ensembling [20]. Let $\mathbf{t}^{[i]}(k)$ and $\mathbf{p}^{[i]}(k)$ denote the target and model output respectively for example $i$ at iteration $k$ of training. We set

$$\mathbf{t}^{[i]}(k) := \beta\mathbf{t}^{[i]}(k-1) + (1-\beta)\mathbf{p}^{[i]}(k), \tag{9}$$

where $0 \leq \beta < 1$ is the momentum. The basic version of our proposed method alternates between computing the targets and minimizing the cost function (6) via stochastic gradient descent.

Target estimation can be further improved in two ways. First, by using the output of a model obtained through a running average of the model weights during training. In semi-supervised learning, this *weight averaging* approach has been proposed to mitigate confirmation bias [40]. Second, by using two separate neural networks, where the target of each network is computed from the output of the other network. The approach is inspired by Co-teaching and related methods [14, 52, 22]. The ablation results in Section 6 show that weight averaging, two networks, and mixup data augmentation [55] all separately improve performance. We call the combination of all these elements ELR+. A detailed description of ELR and ELR+ is provided in Section F of the supplementary material.

## 5 Experiments

We evaluate the proposed methodology on two standard benchmarks with simulated label noise, CIFAR-10 and CIFAR-100 [18], and two real-world datasets, Clothing1M [47] and WebVision [24]. For CIFAR-10 and CIFAR-100 we simulate label noise by randomly flipping a certain fraction of the labels in the training set following a *symmetric* uniform distribution (as in Eq. (1)), as well as a more realistic *asymmetric* class-dependent distribution, following the scheme proposed in [31]. Clothing1M consists of 1 million training images collected from online shopping websites with labels generated using surrounding text. Its noise level is estimated at 38.5% [36]. For ease of comparison to previous works [17, 7], we consider the mini WebVision dataset which contains the top 50 classes from the Google image subset of WebVision, which results in approximately 66 thousand images. The noise level of WebVision is estimated at 20% [24]. Table G.1 in the supplementary material reports additional details about the datasets, and our training, validation and test splits.

In our experiments, we prioritize making our results comparable to the existing literature. When possible we use the same preprocessing, and architectures as previous methods. The details are described in Section G of the supplementary material. We focus on two variants of the proposed approach: ELR with temporal ensembling, which we call ELR, and ELR with temporal ensembling, weight averaging, two networks, and mixup data augmentation, which we call ELR+ (see Section F). The choice of hyperparameters is performed on separate validation sets. Section H shows that the sensitivity to different hyperparameters is quite low. Finally, we also perform an ablation study on CIFAR-10 for two levels of symmetric noise (40% and 80%) in order to evaluate the contribution of

| | | | Cross entropy | Co-teaching+ [52] | Mixup [55] | PENCIL [51] | MD-DYR-SH [2] | DivideMix [22] | ELR+ | ELR+* |
|---|---|---|---|---|---|---|---|---|---|---|
| CIFAR-10 | Sym. label noise | 20% | 86.8 | 89.5 | 95.6 | 92.4 | 94.0 | **96.1** | 94.6 | 95.8 |
| | | 50% | 79.4 | 85.7 | 87.1 | 89.1 | 92.0 | 94.6 | 93.8 | **94.8** |
| | | 80% | 62.9 | 67.4 | 71.6 | 77.5 | 86.8 | 93.2 | 91.1 | **93.3** |
| | | 90% | 42.7 | 47.9 | 52.2 | 58.9 | 69.1 | 76.0 | 75.2 | **78.7** |
| | Asym. | 40% | 83.2 | - | - | 88.5 | 87.4 | **93.4** | 92.7 | 93.0 |
| CIFAR-100 | Sym. label noise | 20% | 62.0 | 65.6 | 67.8 | 69.4 | 73.9 | 77.3 | 77.5 | **77.6** |
| | | 50% | 46.7 | 51.8 | 57.3 | 57.5 | 66.1 | **74.6** | 72.4 | 73.6 |
| | | 80% | 19.9 | 27.9 | 30.8 | 31.1 | 48.2 | 60.2 | 58.2 | **60.8** |
| | | 90% | 10.1 | 13.7 | 14.6 | 15.3 | 24.3 | 31.5 | 30.8 | **33.4** |
| | Asym. | 40% | - | - | - | - | - | 72.1 | 76.5 | **77.5** |

Table 2: Comparison with state-of-the-art methods on CIFAR-10 and CIFAR-100 with symmetric and asymmetric noise. For ELR+, we use 10% of the training set for validation, and treat the validation set as a held-out test set. The result for DivideMix on CIFAR-100 with 40% asymmetric noise was obtained using publicly available code. The rest of the results are taken from [22], which reports the highest accuracy observed on the validation set during training. We also report the performance of ELR+ under this metric on the rightmost column (ELR+*).

the different elements in ELR+. Code to reproduce the experiments is publicly available online at https://github.com/shengliu66/ELR.

# 6 Results

Table 1 evaluates the performance of ELR on CIFAR-10 and CIFAR-100 with different levels of symmetric and asymmetric label noise. We compare to the best performing methods that only modify the training loss. All techniques use the same architecture (ResNet34), batch size, and training procedure. ELR consistently outperforms the rest by a significant margin. To illustrate the influence of the training procedure, we include results with a different learning-rate scheduler (cosine annealing [26]), which further improves the results.

In Table 2, we compare ELR+ to state-of-the-art methods, which also apply sample selection and data augmentation, on CIFAR-10 and CIFAR-100. All methods use the same architecture (**PreAct ResNet-18**). The results from other methods may not be completely comparable to ours because they correspond to the best test performance during training, whereas we use a separate validation set. Nevertheless, ELR+ outperforms all other methods except DivideMix.

| CE | Forward [31] | GCE [56] | SL [45] | Joint-Optim [38] | DivideMix [22] | ELR | ELR+ |
|---|---|---|---|---|---|---|---|
| 69.10 | 69.84 | 69.75 | 71.02 | 72.16 | 74.76 | 72.87 | **74.81** |

Table 3: Comparison with state-of-the-art methods in test accuracy (%) on Clothing1M. All methods use a ResNet-50 architecture pretrained on ImageNet. Results of other methods are taken from the original papers (except for GCE, which is taken from [45]).

Table 3 compares ELR and ELR+ to state-of-the-art methods on the Clothing1M dataset. ELR+ achieves state-of-the-art performance, slightly superior to DivideMix.

| | | D2L [27] | MentorNet [17] | Co-teaching [14] | Iterative-CV [44] | DivideMix [22] | ELR | ELR+ |
|---|---|---|---|---|---|---|---|---|
| WebVision | top1 | 62.68 | 63.00 | 63.58 | 65.24 | 77.32 | 76.26 | **77.78** |
| | top5 | 84.00 | 81.40 | 85.20 | 85.34 | 91.64 | 91.26 | **91.68** |
| ILSVRC12 | top1 | 57.80 | 57.80 | 61.48 | 61.60 | **75.20** | 68.71 | 70.29 |
| | top5 | 81.36 | 79.92 | 84.70 | 84.98 | **90.84** | 87.84 | 89.76 |

Table 4: Comparison with state-of-the-art methods trained on the mini WebVision dataset. Results of other methods are taken from [22]. All methods use an InceptionResNetV2 architecture.

Table 4 compares ELR and ELR+ to state-of-the-art methods trained on the mini WebVision dataset and evaluated on both the WebVision and ImageNet ILSVRC12 validation sets. ELR+ achieves state-of-the-art performance, slightly superior to DivideMix, on WebVision. ELR also performs strongly, despite its simplicity. On ILSVRC12 DivideMix produces superior results (particularly in terms of top1 accuracy).

|  |  |  | 40% | | 80% | |
|---|---|---|---|---|---|---|
|  |  |  | Weight Averaging | | Weight Averaging | |
|  |  |  | ✓ | ✗ | ✓ | ✗ |
| 1 Network | mixup | ✓ | $93.04 \pm 0.12$ | $91.05 \pm 0.13$ | $87.23 \pm 0.30$ | $81.43 \pm 0.52$ |
|  |  | ✗ | $92.09 \pm 0.08$ | $90.83 \pm 0.07$ | $76.50 \pm 0.65$ | $72.54 \pm 0.35$ |
| 2 Networks | mixup | ✓ | $93.68 \pm 0.51$ | $93.51 \pm 0.47$ | $88.62 \pm 0.26$ | $84.75 \pm 0.26$ |
|  |  | ✗ | $92.95 \pm 0.05$ | $91.86 \pm 0.14$ | $80.13 \pm 0.51$ | $73.49 \pm 0.47$ |

Table 5: Ablation study evaluating the influence of weight averaging, the use of two networks, and mixup data augmentation for the CIFAR-10 dataset with medium (40%) and high (80%) levels of symmetric noise. The mean accuracy and its standard deviation are computed over five noise realizations.

Table 5 shows the results of an ablation study evaluating the influence of the different elements of ELR+ for the CIFAR-10 dataset with medium (40%) and high (80%) levels of symmetric noise. Each element seems to provide an independent performance boost. At the medium noise level the improvement is modest, but at the high noise level it is very significant. This is in line with recent works showing the effectiveness of semi-supervised learning techniques in such settings [2, 22].

## 7    Discussion and Future Work

In this work we provide a theoretical characterization of the early-learning and memorization phenomena for a linear generative model, and build upon the resulting insights to propose a novel framework for learning from data with noisy annotations. Our proposed methodology yields strong results on standard benchmarks and real-world datasets for several different network architectures. However, there remain multiple open problems for future research. On the theoretical front, it would be interesting to bridge the gap between linear and nonlinear models (see [23] for some work in this direction), and also to investigate the dynamics of the proposed regularization scheme. On the methodological front, we hope that our work will trigger interest in the design of new forms of regularization that provide robustness to label noise.

## 8    Broader Impact

This work has the potential to advance the development of machine-learning methods that can be deployed in contexts where it is costly to gather accurate annotations. This is an important issue in applications such as medicine, where machine learning has great potential societal impact.

**Acknowledgments**

This research was supported by NSF NRT-HDR Award 1922658. SL was partially supported by NSF grant DMS 2009752. CFG was partially supported by NSF Award HDR-1940097. JNW gratefully acknowledges the support of the Institute for Advanced Study, where a portion of this research was conducted.

## Footnotes

[1]The term target is inspired by semi-supervised learning where target probabilities are used to learn on unlabeled examples [50, 29, 20].

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
