[Supplementary Material]

# A Theoretical analysis of early learning and memorization in a linear model

In this section, we formalize and substantiate the claims of Theorem 1.

Theorem 1 has three parts, which we address in the following sections. First, in Section A.2, we show that the classifier makes progress during the early-learning phase: over the first $T$ iterations, the gradient is well correlated with $\mathbf{v}$ and the accuracy on mislabeled examples increases. However, as noted in the main text, this early progress halts because the gradient terms corresponding to correctly labeled examples begin to disappear. We prove this rigorously in Section A.3, which shows that the overall magnitude of the gradient terms corresponding to correctly labeled examples shrinks over the first $T$ iterations. Finally, in Section A.4, we prove the claimed asymptotic behavior: as $t \to \infty$, gradient descent perfectly memorizes the noisy labels.

## A.1 Notation and setup

We consider a softmax regression model parameterized by two weight vectors $\Theta_1$ and $\Theta_2$, which are the rows of the parameter matrix $\Theta \in \mathbb{R}^{2 \times p}$. In the linear case this is equivalent to a logistic regression model, because the cross-entropy loss on two classes depends only on the vector $\Theta_1 - \Theta_2$. If we reparametrize the labels as

$$\varepsilon^{[i]} = \begin{cases} 1 & \text{if } \mathbf{y}_1^{[i]} = 1 \\ -1 & \text{if } \mathbf{y}_2^{[i]} = 1 \,, \end{cases}$$

and set $\theta := \Theta_1 - \Theta_2$, we can then write the loss as

$$\mathcal{L}_{\text{CE}}(\theta) = \frac{1}{n} \sum_{i=1}^{n} \log(1 + e^{-\varepsilon^{[i]} \theta^\top \mathbf{x}^{[i]}}) \,.$$

We write $\varepsilon^*$ for the true cluster assignments: $(\varepsilon^*)^{[i]} = 1$ if $\mathbf{x}^{[i]}$ comes from the cluster with mean $+\mathbf{v}$, and $(\varepsilon^*)^{[i]} = -1$ otherwise. Note that, with this convention, we can always write $\mathbf{x}^{[i]} = (\varepsilon^*)^{[i]}(\mathbf{v} - \sigma \mathbf{z}^{[i]})$, where $\mathbf{z}^{[i]}$ is a standard Gaussian random vector independent of all other random variables.

In terms of $\theta$ and $\varepsilon$, the gradient (2) reads

$$\nabla \mathcal{L}_{\text{CE}}(\theta) = \frac{1}{2n} \sum_{i=1}^{n} \mathbf{x}^{[i]} \left( \tanh(\theta^\top \mathbf{x}^{[i]}) - \varepsilon^{[i]} \right) , \tag{10}$$

As noted in the main text, the coefficient $\tanh(\theta^\top \mathbf{x}^{[i]}) - \varepsilon^{[i]}$ is the key quantity governing the properties of the gradient.

Let us write $C$ for the set of indices for which the labels are correct, and $W$ for the set of indices for which labels are wrong.

We assume that $\theta_0$ is initialized randomly on the sphere with radius 2, and then optimized to minimize $\mathcal{L}$ via gradient descent with fixed step size $\eta < 1$. We denote the iterates by $\theta_t$.

We consider the asymptotic regime where $\sigma \ll 1$ and $\Delta$ are constants and $p, n \to \infty$, with $p/n \in (1 - \Delta/2, 1)$. We will let $\sigma_\Delta$ denote a constant, whose precise value may change from proposition to proposition; however, in all cases the requirements on $\sigma$ will be *independent* of $p$ and $n$. For convenience, we assume that $\Delta \leq 1/2$, though it is straightforward to extend the analysis below to any $\Delta$ bounded away from 1. Note that when $\Delta = 1$, each observed label is independent of the data, so no learning is possible. We will use the phrase "with high probability" to denote an event which happens with probability $1 - o(1)$ as $n, p \to \infty$, and we use $o_P(1)$ to denote a random quantity which converges to 0 in probability. We use the symbol $c$ to refer to an unspecified positive constant whose value may change from line to line. We use subscripts to indicate when this constant depends on other parameters of the problem.

We let $T$ be the smallest positive integer such that $\theta_T^\top \mathbf{v} \geq 1/10$. By Lemmas 7 and 8 in Section A.5, $T = \Omega(1/\eta)$ with high probability.

## A.2 Early-learning succeeds

We first show that, for the first $T$ iterations, the negative gradient $-\nabla \mathcal{L}_{\text{CE}}(\theta_t)^\top$ has constant correlation with $\mathbf{v}$. (Note that, by contrast, a *random* vector in $\mathbb{R}^p$ typically has *negligible* correlation with $\mathbf{v}$.)

**Proposition 3.** *There exists a constant $\sigma_\Delta$, depending only on $\Delta$, such that if $\sigma \leq \sigma_\Delta$ then with high probability, for all $t < T$, we have $\|\theta_t - \theta_0\| \leq 1$ and*

$$-\nabla \mathcal{L}_{CE}(\theta_t)^\top \mathbf{v}/\|\nabla \mathcal{L}_{CE}(\theta_t)\| \geq 1/6 \,.$$

*Proof.* We will prove the claim by induction. We write

$$-\nabla\mathcal{L}_{\mathrm{CE}}(\theta_t) = \frac{1}{2n}\sum_{i=1}^{n}\varepsilon^{[i]}\mathbf{x}^{[i]} - \frac{1}{2n}\sum_{i=1}^{n}\mathbf{x}^{[i]}\tanh(\theta_t^{\top}\mathbf{x}^{[i]})\,.$$

Since $\mathbb{E}\mathbf{v}^{\top}(\varepsilon^{[i]}\mathbf{x}^{[i]}) = (1-\Delta)$, the law of large numbers implies

$$\mathbf{v}^{\top}\Big(\frac{1}{2n}\sum_{i=1}^{n}\varepsilon^{[i]}\mathbf{x}^{[i]}\Big) = \frac{1}{2}(1-\Delta) + o_P(1)\,.$$

Moreover, by Lemma 9, there exists a positive constant $c$ such that with high probability

$$\Big|\mathbf{v}^{\top}\Big(\frac{1}{2n}\sum_{i=1}^{n}\mathbf{x}^{[i]}\tanh(\theta_t^{\top}\mathbf{x}^{[i]})\Big)\Big| \le \frac{1}{2}\Big(\frac{1}{n}\sum_{i=1}^{n}(\mathbf{v}^{\top}\mathbf{x}^{[i]})^2\Big)^{1/2}\Big(\frac{1}{n}\sum_{i=1}^{n}\tanh(\theta_t^{\top}\mathbf{x}^{[i]})^2\Big)^{1/2}$$

$$\le \frac{1}{2}|\tanh(\theta_t^{\top}\mathbf{v})| + c\sigma(1 + \|\theta_t - \theta_0\|)\,.$$

Thus, applying Lemma 8 yields that with high probability

$$-\nabla\mathcal{L}_{\mathrm{CE}}(\theta_t)^{\top}v/\|\nabla\mathcal{L}_{\mathrm{CE}}(\theta_t)\| \ge \frac{1}{2}((1-\Delta) - |\tanh(\theta_t^{\top}\mathbf{v})|) - c\sigma(1 + \|\theta_t - \theta_0\|)\,. \tag{11}$$

When $t = 0$, the first term is $\frac{1}{2}(1-\Delta) + o_P(1)$ by Lemma 7, and the second term is $c\sigma$. Since we have assumed that $\Delta \le 1/2$, as long as $\sigma \le \sigma_\Delta < c^{-1}\left(\frac{2}{3} - \frac{\Delta}{2}\right)$ we will have that $-\nabla\mathcal{L}_{\mathrm{CE}}(\theta_0)^{\top}v/\|\nabla\mathcal{L}_{\mathrm{CE}}(\theta_0)\| \ge 1/6$ with high probability, as desired.

We proceed with the induction. We will show that $\|\theta_t - \theta_0\| \le 1$ with high probability for $t < T$, and use (11) to show that this implies the desired bound on the correlation of the gradient. If we assume the claim holds up to time $t$, then the definition of gradient descent implies

$$\theta_t - \theta_0 = \eta\sum_{s=0}^{t-1}\mathbf{g}_s\,,$$

where $\mathbf{g}_s$ satisfies $\mathbf{g}_s^{\top}\mathbf{v}/\|\mathbf{g}_s\| \ge 1/6$. Since the set of vectors satisfying this requirement forms a convex cone, we obtain that

$$(\theta_t - \theta_0)^{\top}\mathbf{v}/\|\theta_t - \theta_0\| \ge 1/6$$

From this observation, we obtain two facts about $\theta_t$. First, since $t < T$, the definition of $T$ implies that $\theta_t^{\top}\mathbf{v} < .1$. Since $|\theta_0^{\top}\mathbf{v}| = o_P(1)$ by Lemma 7, we obtain that $\|\theta_t - \theta_0\| \le 1$ with high probability. Second, $\theta_t^{\top}\mathbf{v} \ge \theta_0^{\top}\mathbf{v}$, and since $|\theta_0^{\top}\mathbf{v}| = o_P(1)$ we have in particular that $\theta_t^{\top}\mathbf{v} > -.1$. Therefore, with high probability, we also have $|\theta_t\top\mathbf{v}| < .1$

Examining (11), we therefore see that the quantity on the right side is at least

$$\frac{1}{2}((1-\Delta) - .1) - 2c\sigma\,.$$

Again since we have assumed that $\Delta \le 1/2$, as long as $\sigma \le \sigma_\Delta < (2c)^{-1}\left(\frac{2}{3} - \frac{\Delta}{2} - .1\right)$, we obtain by (11) that $-\nabla\mathcal{L}_{\mathrm{CE}}(\theta_t)^{\top}v/\|\nabla\mathcal{L}_{\mathrm{CE}}(\theta_t)\| \ge 1/6$.

$\square$

Given $\theta_t$, we denote by

$$\hat{\mathcal{A}}(\theta_t) = \frac{1}{|W|}\sum_{i \in W}\mathbb{1}\{\mathrm{sign}(\theta_t^{\top}\mathbf{x}^{[i]}) = (\varepsilon^*)^{[i]}\}$$

the accuracy of $\theta_t$ on mislabeled examples. We now show that the classifier's accuracy on the mislabeled examples improves over the first $T$ rounds. In fact, we show that with high probability, $\hat{\mathcal{A}}(\theta_0) \approx 1/2$ whereas $\hat{\mathcal{A}}(\theta_T) \approx 1$.

**Theorem 4.** *There exists a $\sigma_\Delta$ such that if $\sigma \le \sigma_\Delta$, then*

$$\hat{\mathcal{A}}(\theta_0) \le .5001 \tag{12}$$

*and*

$$\hat{\mathcal{A}}(\theta_T) > .9999 \tag{13}$$

*with high probability.*

*Proof.* Let us write $\mathbf{x}^{[i]} = (\varepsilon^*)^{[i]}(\mathbf{v} - \sigma\mathbf{z}^{[i]})$, where $\mathbf{z}^{[i]}$ is a standard Gaussian vector. If we fix $\theta_0$, then $\operatorname{sign}(\theta_0^\top \mathbf{x}^{[i]}) = (\varepsilon^*)^{[i]}$ if and only if $\sigma\theta_0^\top\mathbf{z}^{[i]} < \theta_0^\top\mathbf{v}$. In particular this yields

$$\mathbb{E}[\mathbb{1}\{\operatorname{sign}(\theta_0^\top\mathbf{x}^{[i]}) = (\varepsilon^*)^{[i]}\}|\theta_0] = \mathbb{P}[\sigma\theta_0^\top\mathbf{z}^{[i]} < \theta_0^\top\mathbf{v}|\theta_0] \leq 1/2 + O(|\theta_0^\top\mathbf{v}|/\sigma)\,.$$

By the law of large numbers, we have that, conditioned on $\theta_0$,

$$\hat{\mathcal{A}}(\theta_0) \leq 1/2 + O(|\theta_0^\top\mathbf{v}|/\sigma) + o_P(1)\,,$$

and applying Lemma 7 yields $\hat{\mathcal{A}}(\theta_0) \leq 1/2 + o_P(1)$.

In the other direction, we employ a method based on [31]. The proof of Proposition 3 establishes that $\|\theta_t - \theta_0\| \leq 1$ for all $t < T - 1$ with high probability. Since $\eta < 1$ and $\|\theta_0\| = 2$, Lemma 8 implies that as long as $\sigma < 1/2$, $\|\theta_T\| \leq 5$ with high probability. Since $\theta_T^\top\mathbf{v} \geq .1$ by assumption, we obtain that $\theta_T^\top\mathbf{v}/\|\theta_T\| \geq 1/50$ with high probability.

Note that $W$ is a random subset of $[n]$. For now, let us condition on this random variable. If we write $\Phi$ for the Gaussian CDF, then by the same reasoning as above, for any fixed $\theta \in \mathbb{R}^p$,

$$\mathbb{E}[\mathbb{1}\{\operatorname{sign}(\theta^\top\mathbf{x}^{[i]}) = (\varepsilon^*)^{[i]}\}] = \mathbb{P}[\sigma\theta^\top\mathbf{z}^{[i]} < \theta^\top\mathbf{v}] = \Phi(\sigma^{-1}\theta^\top\mathbf{v}/\|\theta\|)$$

Therefore, if $\theta^\top\mathbf{v}/\|\theta\| \geq \tau$, then for any $\delta > 0$, we have

$$\hat{\mathcal{A}}(\theta) \geq \Phi(\sigma^{-1}\tau - \delta) - \frac{1}{|W|}\sum_{i \in W} \Phi(\sigma^{-1}\tau - \delta) - \mathbb{1}\{\theta^\top\mathbf{z}^{[i]}/\|\theta\| < \sigma^{-1}\tau\} \tag{14}$$

Set

$$\phi(x) := \begin{cases} 1 & \text{if } x < \sigma^{-1}\tau - \delta \\ \frac{1}{\delta}(\sigma^{-1}\tau - x) & \text{if } x \in [\sigma^{-1}\tau - \delta, \sigma^{-1}\tau] \\ 0 & \text{if } x > \sigma^{-1}\tau. \end{cases}$$

By construction, $\phi$ is $\frac{1}{\delta}$-Lipschitz and satisfies

$$\mathbb{1}\{x < \sigma^{-1}\tau - \delta\} \leq \phi(x) \leq \mathbb{1}\{x < \sigma^{-1}\tau\}$$

for all $x \in \mathbb{R}$. In particular, we have

$$\Phi(\sigma^{-1}\tau - \delta) - \mathbb{1}\{\theta^\top\mathbf{z}^{[i]}/\|\theta\| < \sigma^{-1}\tau\} \leq \mathbb{E}[\phi(\theta^\top\mathbf{z}^{[i]}/\|\theta\|)] - \phi(\theta^\top\mathbf{z}^{[i]}/\|\theta\|)\,.$$

Denote the set of $\theta \in \mathbb{R}^p$ satisfying $\theta^\top\mathbf{v}/\|\theta\| \geq \tau$ by $\mathcal{C}_\tau$. Combining the last display with (14) yields

$$\mathbb{E}\inf_{\theta \in \mathcal{C}_\tau}\hat{\mathcal{A}}(\theta) \geq \Phi(\sigma^{-1}\tau - \delta) - \mathbb{E}\sup_{\theta \in \mathcal{C}_\tau}\frac{1}{|W|}\sum_{i \in W}\mathbb{E}[\phi(\theta^\top\mathbf{z}^{[i]}/\|\theta\|)] - \phi(\theta^\top\mathbf{z}^{[i]}/\|\theta\|)\,.$$

To control the last term, we employ symmetrization and contraction (see [21, Chapter 4]) to obtain

$$\mathbb{E}\sup_{\theta \in \mathcal{C}_\tau}\frac{1}{|W|}\sum_{i \in W}\mathbb{E}[\phi(\theta^\top\mathbf{z}^{[i]}/\|\theta\|)] - \phi(\theta^\top\mathbf{z}^{[i]}/\|\theta\|) \leq \mathbb{E}\sup_{\theta \in \mathcal{C}_\tau}\frac{1}{|W|}\sum_{i \in W}\epsilon_i\phi(\theta^\top\mathbf{z}^{[i]}/\|\theta\|)$$

$$\leq \frac{1}{\delta}\mathbb{E}\sup_{\theta \in \mathcal{C}_\tau}\frac{1}{|W|}\sum_{i \in W}\epsilon_i\theta^\top\mathbf{z}^{[i]}/\|\theta\|$$

$$\leq \frac{1}{\delta}\mathbb{E}\sup_{\theta \in \mathbb{R}^p}\frac{1}{|W|}\sum_{i \in W}\epsilon_i\theta^\top\mathbf{z}^{[i]}/\|\theta\|$$

$$= \frac{1}{\delta}\mathbb{E}\left\|\frac{1}{|W|}\sum_{i \in W}\epsilon_i\mathbf{z}^{[i]}\right\|\,.$$

where $\epsilon_i$ are independent Rademacher random variables. The final quantity is easily seen to be at most $\frac{1}{\delta}\sqrt{p/|W|}$. Therefore we have

$$\mathbb{E}\inf_{\theta \in \mathcal{C}_\tau}\hat{\mathcal{A}}(\theta) \geq \Phi(\sigma^{-1}\tau - \delta) - \frac{1}{\delta}\sqrt{p/|W|}\,,$$

and a standard application of Azuma's inequality implies that this bound also holds with high probability. Since $\theta_T^\top\mathbf{v}/\|\theta_T\| \geq 1/50$ and $|W| \geq \Delta n/2$ with high probability, there exists a positive constant $c_\Delta$ such that

$$\hat{\mathcal{A}}(\theta_T) \geq \Phi((50\sigma)^{-1} - \delta) - c_\Delta/\delta\,.$$

If we choose $\delta = 10^{-4}/2c_\Delta$, then there exists a $\sigma_\Delta$ for which $\Phi((50\sigma_\Delta)^{-1} - \delta) > 1 - 10^{-4}/2$. We obtain that for any $\sigma \leq \sigma_\Delta$, $\hat{\mathcal{A}}(\theta_T) \leq 1 - 10^{-4}$, as claimed. $\qquad\square$

### A.3 Vanishing gradients

We now show that, over the first $T$ iterations, the coefficients $\tanh(\theta^\top \mathbf{x}^{[i]}) - \varepsilon^{[i]}$ associated with the correctly labeled examples decrease, while the coefficients on mislabeled examples increase. For simplicity, we write $\kappa^{[i]} := \tanh(\theta^\top \mathbf{x}^{[i]}) - \varepsilon^{[i]}$.

**Proposition 5.** *There exists a constant $\sigma_\Delta$ such that, for any $\sigma \leq \sigma_\Delta$, with high probability,*

$$\frac{1}{|C|} \sum_{i \in C} (\kappa^{[i]}(\theta_T))^2 < \frac{1}{|C|} \sum_{i \in C} (\kappa^{[i]}(\theta_0))^2 - .05$$

$$\frac{1}{|W|} \sum_{i \in W} (\kappa^{[i]}(\theta_T))^2 > \frac{1}{|W|} \sum_{i \in W} (\kappa^{[i]}(\theta_0))^2 + .05 \,.$$

*That is, during the first stage, the coefficients on correct examples decrease while the coefficients on wrongly labeled examples increase.*

*Proof.* Let us first consider

$$\frac{1}{|C|} \sum_{i \in C} (\tanh(\theta_0^\top \mathbf{x}^{[i]}) - \varepsilon^{[i]})^2$$

For fixed initialization $\theta_0$, the law of large numbers implies that this quantity is near

$$\mathbb{E}_{\mathbf{x},\varepsilon}(\varepsilon \tanh(\theta_0^\top \mathbf{x}) - 1)^2 \geq \left( \mathbb{E}_{\mathbf{x},\varepsilon} \varepsilon \tanh(\theta_0^\top \mathbf{x}) - 1 \right)^2 .$$

Let us write $\mathbf{x} = \varepsilon^*(\mathbf{v} - \sigma\mathbf{z})$, where $\mathbf{z}$ is a standard Gaussian vector. Then the fact that $\tanh$ is Lipschitz implies

$$\mathbb{E}_{\mathbf{x},\varepsilon} \varepsilon \tanh(\theta_0^\top \mathbf{x}) \leq \mathbb{E}_{\mathbf{x},\varepsilon} \varepsilon \tanh(\varepsilon^* \sigma \theta_0^\top \mathbf{z}) + |\theta_0^\top \mathbf{v}| = |\theta_0^\top \mathbf{v}| \,,$$

where we have used that $\mathbb{E}[\tanh(\varepsilon^* \sigma \theta_0^\top \mathbf{z})|\varepsilon] = 0$. By Lemma 7, $|\theta_0^\top \mathbf{v}| = o_P(1)$. Hence

$$\frac{1}{|C|} \sum_{i \in C} (\tanh(\theta_0^\top \mathbf{x}^{[i]}) - \varepsilon^{[i]})^2 \geq 1 - o_P(1) \,.$$

At iteration $T$, we have that $\theta_T^\top \mathbf{v} \geq .1$, by assumption, and $\|\theta_T - \theta_0\| \leq 3$, by the proof of Proposition 3. We can therefore apply Lemma 9 to obtain

$$\left( \frac{1}{|C|} \sum_{i \in C} (\kappa^{[i]})^2 \right)^{1/2} \leq \left( \frac{1}{|C|} \sum_{i \in C} ((\varepsilon^*)^{[i]} \tanh(\theta_T^\top \mathbf{v}) - \varepsilon^{[i]}) \right)^{1/2} + \sigma(2 + 3c_\Delta) + o_P(1)$$

$$= |\tanh(\theta_T^\top \mathbf{v}) - 1| + + \sigma(2 + 3c_\Delta) + o_P(1)$$

$$\leq |\tanh(.1) - 1| + \sigma(2 + 3c_\Delta) + o_P(1) \,,$$

where the equality uses the fact that $(\varepsilon^*)^{[i]} = \varepsilon^{[i]}$ for all $i \in C$. As long as $\sigma \leq \sigma_\Delta < .01/(2 + 3c_\Delta)$ this quantity is strictly less than .95. We therefore obtain that, for $\sigma \leq \sigma_\Delta$, $\frac{1}{|C|} \sum_{i \in C} (\kappa^{[i]}(\theta_T))^2 < \frac{1}{|C|} \sum_{i \in C} (\kappa^{[i]}(\theta_0))^2 - .05$ with high probability. This proves the first claim.

The second claim is established by an analogous argument: for fixed initialization $\theta_0$, we have

$$\mathbb{E} \tanh(\theta_0^\top \mathbf{x})^2 \leq \mathbb{E}(\theta_0^\top \mathbf{x})^2 = 4\sigma^2 + (\theta_0^\top \mathbf{v})^2 \,,$$

so as above we can conclude that

$$\frac{1}{|W|} \sum_{i \in W} (\kappa^{[i]}(\theta_0))^2 \leq 1 + 4\sigma^2 + o_P(1) \,.$$

We likewise have by another application of Lemma 9

$$\left( \frac{1}{|W|} \sum_{i \in W} (\tanh(\theta_T^\top \mathbf{x}^{[i]}) - \varepsilon^{[i]})^2 \right)^{1/2} \geq |\tanh(-\theta_T^\top \mathbf{v}) - 1| - \sigma(2 + 3c_\Delta) - o_P(1)$$

$$\geq 1 + \tanh(.1) - \sigma(2 + 3c_\Delta) - o_P(1) \,.$$

where we again have used that $(\varepsilon^*)^{[i]} = -\varepsilon^{[i]}$ for $i \in W$. If we choose $\sigma_\Delta$ small enough that

$$1.05 + 4\sigma_\Delta^2 < (1 + \tanh(.1) - \sigma_\Delta(2 + 3c_\Delta))^2 \,,$$

then we will have for all $\sigma \leq \sigma_\Delta$ that $\frac{1}{|W|} \sum_{i \in W} (\kappa^{[i]}(\theta_T))^2 > \frac{1}{|W|} \sum_{i \in W} (\kappa^{[i]}(\theta_0))^2 + .05$ with high probability. This proves the claim. $\square$

## A.4 Memorization

To show that the labels are memorized asymptotically, it suffices to show that the classes $S_+ := \{\mathbf{x}^{[i]} : \varepsilon^{[i]} = +1\}$ and $S_- := \{\mathbf{x}^{[i]} : \varepsilon^{[i]} = -1\}$ are linearly separable. Indeed, it is well known that for linearly separable data, gradient descent performed on the logistic loss will yield a classifier which perfectly memorizes the labels [see, e.g. 38, Lemma 1]. It is therefore enough to establish the following theorem.

**Theorem 6.** *If $p, n \to \infty$ and $\liminf_{p,n\to\infty} p/n > 1 - \Delta/2$, then the classes $S_+$ and $S_-$ are linearly separable with probability tending to $1$.*

*Proof.* Write $X = \{\mathbf{x}^{[1]}, \ldots, \mathbf{x}^{[n]}\}$. Since the samples $\mathbf{x}^{[1]}, \ldots, \mathbf{x}^{[n]}$ are drawn from a distribution absolutely continuous with respect to the Lebesgue measure, they are in general position with probability 1. By a theorem of Schläfli [see 8, Theorem 1], there exist

$$C(n, p) := 2 \sum_{k=0}^{p-1} \binom{n-1}{k}$$

different subsets $S \subseteq X$ that are linearly separable from their complements. In particular, there are at most

$$2^n - C(n, p) = 2 \sum_{k=p}^{n-1} \binom{n-1}{k} = 2 \sum_{k=0}^{n-p-1} \binom{n-1}{k}$$

partitions of $X$ which are *not* separable. Write $B$ for the bad set of non-separable subsets $S \subseteq X$. Conditional on $X$, the probability that the classes $S_+$ and $S_-$ are not separable is just $\mathbb{P}[S_+ \in B | X]$.

Let us write $T_+ := \{\mathbf{x}^{[i]} : (\varepsilon^*)^{[i]} = +1\}$. For each $i$, the example $\mathbf{x}^{[i]}$ is in $S_+$ with probability $1 - (\Delta/2)$ if $i \in T_+$ or $\Delta/2$ or if $i \notin T_+$. We therefore have for any $S \subset X$ that

$$\mathbb{P}[S_+ = S | X] = (\Delta/2)^{|T_+ \triangle S|} (1 - (\Delta/2))^{n - |T_+ \triangle S|}.$$

We obtain that

$$\mathbb{P}[S_+ \in B | X] = \sum_{S \in B} (\Delta/2)^{|T_+ \triangle S|} (1 - (\Delta/2))^{n - |T_+ \triangle S|}$$

$$= \sum_{k=0}^{n} |\{S \in B : |T_+ \triangle S| = k\}| \cdot (\Delta/2)^k (1 - (\Delta/2))^{n-k}. \tag{15}$$

The set $\{S \in B : |T_+ \triangle S| = k\}$ has cardinality at most $\binom{n}{k}$. Moreover, $\sum_{k=0}^{n} |\{S \in B : |T_+ \triangle S| = k\}| = |B|$. We can therefore bound the sum (15) by the following optimization problem:

$$\max_{x_1, \ldots x_n} \sum_{k=0}^{n} x_k \cdot (\Delta/2)^k (1 - (\Delta/2))^{n-k} \tag{16}$$

$$\text{s.t. } x_k \in \left[0, \binom{n}{k}\right], \sum_{k=0}^{n} x_k = |B|.$$

Since $\Delta \le 1$, the probability $(\Delta/2)^k (1 - (\Delta/2))^{n-k}$ is a nonincreasing function of $k$. Therefore, because $|B| \le 2 \sum_{k=0}^{n-p-1} \binom{n-1}{k} \le 2 \sum_{k=0}^{n-p} \binom{n}{k}$, the value of (16) is less than

$$2 \sum_{k=0}^{n-p} \binom{n}{k} (\Delta/2)^k (1 - (\Delta/2))^{n-k} = 2 \cdot \mathbb{P}[\text{Bin}(n, \Delta/2) \le (n - p)].$$

If $\limsup_{n,p\to\infty} 1 - p/n < \Delta/2$, then this probability approaches 0 by the law of large numbers. We have shown that if $\liminf_{n,p} p/n > 1 - \Delta/2$, then

$$\mathbb{P}[S_+ \in B | X] \le 2 \cdot \mathbb{P}[\text{Bin}(n, \Delta/2) \le (n - p)] = o(1)$$

holds $X$-almost surely, which proves the claim.

$\square$

## A.5 Additional lemmas

**Lemma 7.** *Suppose that $\theta_0$ is initialized randomly on the sphere of radius $2$.*

$$|\theta_0^\top \mathbf{v}| = o_P(1).$$

*Proof.* Without loss of generality, take $\mathbf{v} = \mathbf{e}_1$, the first elementary basis vector. Since the coordinates of $\theta_0$ each have the same marginal distribution and $\|\theta_0\|^2 = 2$ almost surely, we must have $\mathbb{E}|\theta_0^\top \mathbf{e}_1|^2 = 2/p$. The claim follows. $\square$

**Lemma 8.**
$$\sup_{\theta \in \mathbb{R}^d} \|\nabla \mathcal{L}_{CE}(\theta)\| \leq 1 + 2\sigma + o_P(1).$$

*Proof.* Denote by $\alpha$ the vector with entries $\alpha_i = \frac{1}{2\sqrt{n}}[\tanh(\theta^\top \mathbf{x}^{[i]}) - \varepsilon^{[i]}]$. Since $|\tanh(x)| \leq 1$ for all $x \in \mathbb{R}$, we have $\|\alpha\| \leq 1$. Therefore

$$\nabla \mathcal{L}_{\mathrm{CE}}(\theta) = \frac{1}{\sqrt{n}} \sum_{i=1}^n \mathbf{x}^{[i]} \alpha_i \leq \left\| \frac{1}{\sqrt{n}} \mathbf{X} \right\|,$$

where $\mathbf{X} \in \mathbb{R}^{p \times n}$ is a matrix whose columns are given by the vectors $\mathbf{x}^{[i]}$. By Lemma 10, we have

$$\left\| \frac{1}{\sqrt{n}} \mathbf{X} \right\| = \left\| \frac{1}{n} \mathbf{X}\mathbf{X}^\top \right\|^{1/2} \leq 1 + 2\sigma + o_P(1).$$

This yields the claim. $\square$

**Lemma 9.** *Fix an initialization $\theta_0$ satisfying $\|\theta_0\| = 2$. For any $\tau > 0$ and for $I = C$ or $I = W$, we have*

$$\sup_{\theta: \|\theta - \theta_0\| \leq \tau} \left( \frac{1}{|I|} \sum_{i \in I} ((\varepsilon^*)^{[i]} \tanh(\theta^\top \mathbf{x}^{[i]}) - \tanh(\theta^\top \mathbf{v}))^2 \right)^{1/2} \leq \sigma(2 + c_\Delta \tau) + o_P(1).$$

*The same claim holds with $I = [n]$ with $c_\Delta$ replaced by 2.*

*Proof.* Let us write $\mathbf{x}^{[i]} = (\varepsilon^*)^{[i]}(\mathbf{v} - \sigma \mathbf{z}^{[i]})$, where $\mathbf{z}^{[i]}$ is a standard Gaussian vector. Since $\tanh$ is odd and 1-Lipschitz, we have

$$|(\varepsilon^*)^{[i]} \tanh(\theta^\top \mathbf{x}^{[i]}) - \tanh(\theta^\top \mathbf{v})| = |\tanh(\theta^\top \mathbf{v} - \theta^\top \sigma \mathbf{z}^{[i]}) - \tanh(\theta^\top \mathbf{v})| \leq \sigma |\theta^\top \mathbf{z}^{[i]}|.$$

We therefore obtain

$$\left( \frac{1}{|I|} \sum_{i \in I} ((\varepsilon^*)^{[i]} \tanh(\theta^\top \mathbf{x}^{[i]}) - \tanh(\theta^\top \mathbf{v}))^2 \right)^{1/2} \leq \sigma \left( \frac{1}{|I|} \sum_{i \in I} (\theta^\top \mathbf{z}^{[i]})^2 \right)^{1/2}$$

$$\leq \sigma \left( \frac{1}{|I|} \sum_{i \in I} (\theta_0^\top \mathbf{z}^{[i]})^2 \right)^{1/2} + \sigma \left( \frac{1}{|I|} \sum_{i \in I} ((\theta - \theta_0)^\top \mathbf{z}^{[i]})^2 \right)^{1/2}$$

$$\leq \sigma \left( \frac{1}{|I|} \sum_{i \in I} (\theta_0^\top \mathbf{z}^{[i]})^2 \right)^{1/2} + \sigma \|\theta - \theta_0\| \left\| \frac{1}{|I|} \sum_{i \in I} \mathbf{z}^{[i]} (\mathbf{z}^{[i]})^\top \right\|.$$

Taking a supremum over all $\theta$ such that $\|\theta - \theta_0\| \leq \tau$ and applying Lemma 10 yields the claim. $\square$

**Lemma 10.** *Assume $p \leq n$. There exists a positive constant $c_\Delta$ depending on $\Delta$ such that for $I = C$ or $I = W$,*

$$\frac{1}{|I|} \sum_{i \in I} (\theta_0^\top \mathbf{z}^{[i]})^2 \leq 2 + o_P(1)$$

$$\left\| \frac{1}{|I|} \sum_{i \in I} \mathbf{z}^{[i]} (\mathbf{z}^{[i]})^\top \right\|^{1/2} \leq c_\Delta + o_P(1)$$

$$\left\| \frac{1}{|I|} \sum_{i \in I} \mathbf{x}^{[i]} (\mathbf{x}^{[i]})^\top \right\|^{1/2} \leq 1 + \sigma c_\Delta + o_P(1).$$

*Moreover, the same claims hold with $I = [n]$, when $c_\Delta$ can be replaced by 2.*

*Proof.* The first claim follows immediately from the law of large numbers. For the second two claims, we first consider the case where $I = [n]$. Let us write $\mathbf{Z}$ for the matrix whose columns are given by the vectors $\mathbf{z}^{[i]}$. Then

$$\left\| \frac{1}{n} \sum_{i \in [n]} \mathbf{z}^{[i]} (\mathbf{z}^{[i]})^\top \right\|^{1/2} = \left\| \frac{1}{n} \mathbf{Z}\mathbf{Z}^\top \right\|^{1/2} = \left\| \frac{1}{\sqrt{n}} \mathbf{Z} \right\| \leq 1 + \sqrt{p/n} + o_P(1),$$

where the last claim is a consequences of standard bounds for the spectral norm of Gaussian random matrices [see, e.g. [43]]. Since $p \leq n$ by assumption, the claimed bound follows. When $I = C$ or $W$, the same argument applies, except that we condition on the set of indices in $I$, which yields that, conditioned on $I$,

$$\left\| \frac{1}{|I|} \sum_{i \in I} \mathbf{z}^{[i]} (\mathbf{z}^{[i]})^\top \right\|^{1/2} \leq 2\sqrt{n/|I|} + o_P(1) \,.$$

For any $\Delta$, the random variable $|I|$ concentrates around its expectation, which is $c_\Delta n$, for some constant $c_\Delta$.

Finally, to bound $\left\| \frac{1}{|I|} \sum_{i \in I} \mathbf{x}^{[i]} (\mathbf{x}^{[i]})^\top \right\|^{1/2}$, we again let $\mathbf{X}$ be a matrix whose columns are given by $\mathbf{x}^{[i]}$. Then we can write

$$\mathbf{X} = \mathbf{v}(\varepsilon^*)^\top + \sigma \mathbf{Z} \,,$$

where $\mathbf{Z}$ is a Gaussian matrix, as above. Therefore

$$\left\| \frac{1}{n} \sum_{i \in [n]} \mathbf{x}^{[i]} (\mathbf{x}^{[i]})^\top \right\|^{1/2} = \left\| \frac{1}{\sqrt{n}} \mathbf{X} \right\| \leq \frac{1}{\sqrt{n}} \left\| \mathbf{v}(\varepsilon^*)^\top \right\| + \sigma \left\| \frac{1}{\sqrt{n}} \mathbf{Z} \right\| \leq 1 + 2\sigma + o_P(1) \,.$$

The extension to $I = C$ or $W$ is as above. $\qquad\square$

Figure A.1: Results of training a two-class softmax regression model with a traditional cross entropy loss (top row) and the proposed method (bottom row) to perform classification on 50 simulated data drawn from a mixture of two Gaussians in $\mathbb{R}^{100}$ with $\sigma = 0.1$, where 40% of the labels are flipped at random. The plots show the fraction of examples with clean labels predicted correctly (green) and incorrectly (red) for examples with clean labels (left column) and wrong labels (right column). Analogously to the deep-learning model in Figure 1, the linear model trained with cross entropy begins by learning to predict the true labels, but eventually memorizes the examples with wrong labels. Early-learning regularization prevents memorization, allowing the model to continue learning on the examples with clean labels to attain high accuracy on examples with clean and wrong labels.

## B   Early Learning and Memorization in Linear and Deep-Learning Models

In this section we provide a numerical example to illustrate the theory in Section 3, and the similarities between the behavior of linear and deep-learning models. We train the two-class softmax linear regression model described in Section 3 on data drawn from a mixture of two Gaussians in $\mathbb{R}^{100}$, where 40% of the labels are flipped at random. Figure A.1 shows the training accuracy on the training set for examples with clean and false labels. Analogously to the deep-learning model in Figure 1, the linear model trained with cross entropy begins by learning to predict the true labels, but eventually memorizes the examples with wrong labels as predicted by our theory. The figure also shows the results of applying our proposed early-learning regularization technique with temporal ensembling. ELR prevents memorization, allowing the model to continue learning on the examples with clean labels to attain high accuracy on examples with clean and wrong labels.

As explained in Section 4.2, for both linear and deep-learning models the effect of label noise on the gradient of the cross-entropy loss for each example $i$ is restricted to the term $\mathbf{p}^{[i]} - \mathbf{y}^{[i]}$, where $\mathbf{p}^{[i]}$ is the probability example assigned by the model to the example and $\mathbf{y}^{[i]}$ is the corresponding label. Figure B.1 plots this quantity for the linear model described in the previous paragraph and for the deep-learning model from Figure 1. In both cases, the label noise flips the sign of the term on the wrong labels (left column). The magnitude of this term dominates after early learning (right column), eventually producing memorization of the wrong labels.

## C   Regularization Based on Kullback-Leibler Divergence

A natural alternative to our proposed regularization would be to penalize the Kullback-Leibler (KL) divergence between the the model output and the targets. This results in the following loss function

$$\mathcal{L}_{\text{CE}}(\Theta) - \frac{\lambda}{n} \sum_{i=1}^{n} \sum_{c=1}^{C} \mathbf{t}_c^{[i]} \log \mathbf{p}_c^{[i]}. \tag{17}$$

Figure C.1 shows the result of applying this regularization to CIFAR-10 dataset with 40% symmetric noise for different values of the regularization parameter $\lambda$, using targets computed via temporal ensembling. In contrast to ELR, which succeeds in avoiding memorization while allowing the model to learn effectively as demonstrated

Figure B.1: The effect of label noise on the gradient of the cross-entropy loss for each example $i$ is restricted to the term $\mathbf{p}^{[i]} - \mathbf{y}^{[i]}$, where $\mathbf{p}^{[i]}$ is the probability example assigned by the model to the example and $\mathbf{y}^{[i]}$ is the corresponding label (see Section 4.2). The plots show this term (left column) and its magnitude (right column) for the same linear model as in Figure A.1 (top row) and the same ResNet-34 on CIFAR-10 as in Figure 1 (bottom row) with 40% symmetric noise. On the left, we plot the entry of $\mathbf{p}^{[i]} - \mathbf{y}^{[i]}$ corresponding to the true class, denoted by $c^*$, for training examples with clean (blue) and wrong (red) labels. On the right, we plot the absolute value of the entry. During early learning, the clean labels dominate, but afterwards their effects decrease and the noisy labels start to be dominant, eventually leading to memorization of the wrong labels. In all plots the curves represent the mean value, and the shaded regions are within one standard deviation of the mean.

in the bottom row of Figure 1, regularization based on KL divergence fails to provide robustness. When $\lambda$ is small, memorization of the wrong labels leads to overfitting as in cross-entropy minimization. Increasing $\lambda$ delays memorization, but does not eliminate it. Instead, the model starts overfitting the initial estimates, whether correct or incorrect, and then eventually memorizes the wrong labels (see the bottom right graph in Figure C.1).

Analyzing the gradient of the cost function sheds some light on the reason for the failure of this type of regularization. The gradient with respect to the model parameters $\Theta$ equals

$$\frac{1}{n} \sum_{i=1}^{n} \nabla \mathcal{N}_{\mathbf{x}^{[i]}}(\Theta) \left( \left( \mathbf{p}^{[i]} - \mathbf{y}^{[i]} \right) + \lambda \left( \mathbf{p}^{[i]} - \mathbf{t}^{[i]} \right) \right). \tag{18}$$

A key difference between this gradient and the gradient of ELR is the dependence of the sign of the regularization component on the targets. In ELR, the sign of the $c$th entry for the $i$th is determined by the difference between $\mathbf{t}_c^{[i]}$ and the rest of the entries of $\mathbf{t}^{[i]}$ (see Lemma 2). In contrast, for KL divergence it depends on the difference between $\mathbf{t}_c^{[i]}$ and $\mathbf{p}_c^{[i]}$. This results in overfitting the target probabilities. To illustrate this, recall that for examples with clean labels, the cross-entropy term $\mathbf{p}^{[i]} - \mathbf{y}^{[i]}$ tends to vanish after the early-learning stage because $\mathbf{p}^{[i]}$ is very close to $\mathbf{y}^{[i]}$, allowing examples with wrong labels to dominate the gradient. Let $c^*$ denote the true class. When $\mathbf{p}_{c^*}^{[i]}$ (correctly) approaches one, $\mathbf{t}_{c^*}^{[i]}$ will generally tend to be smaller, because $\mathbf{t}^{[i]}$ is obtained by a moving average and therefore tends to be smoother than $\mathbf{p}^{[i]}$. Consequently, the regularization term tends to decrease $\mathbf{p}_{c^*}^{[i]}$. This is exactly the opposite effect than desired. In contrast, ELR tends to keep $\mathbf{p}_{c^*}^{[i]}$ large, as explained in Section 4.2, which allows the model to continue learning on the clean examples.

# D  The Need for Early Learning Regularization

Our proposed framework consists of two components: target estimation and the early-learning regularization term. Figure D.1 shows that the regularization term is critical to avoid memorization. If we just perform target estimation via temporal ensembling while training with a cross-entropy loss, eventually the targets overfit the noisy labels, resulting in decreased accuracy.

Figure C.1: Results of training a ResNet-34 neural network with a traditional cross entropy loss regularized by KL divergence using different coefficients $\lambda$ (showed in different rows) to perform classification on the CIFAR-10 dataset where 40% of the labels are flipped at random. The left column shows the fraction of examples with clean labels that are predicted correctly (green) and incorrectly (blue). The right column shows the fraction of examples with wrong labels that are predicted correctly (green), *memorized* (the prediction equals the wrong label, shown in red), and incorrectly predicted as neither the true nor the labeled class (blue). When $\lambda = 1$, it is analogous to the model trained without any regularization (top row in Figure 1), while when $\lambda$ increases, the fraction of correctly predicted examples decreases, indicating worse performance.

# E Proof of Lemma 2

To ease notation, we ignore the $i$ superscript, setting $\mathbf{p} := \mathbf{p}^{[i]}$ and $\mathbf{t} := \mathbf{t}^{[i]}$. We denote the instance-level ELR by

$$\mathcal{R}(\Theta) := \log\left(1 - \langle \mathbf{p}, \mathbf{t} \rangle\right). \tag{19}$$

The gradient of $\mathcal{R}$ is

$$\nabla \mathcal{R}(\Theta) = \frac{1}{1 - \langle \mathbf{p}, \mathbf{t} \rangle} \nabla\left(1 - \langle \mathbf{p}, \mathbf{t} \rangle\right). \tag{20}$$

We express the probability estimate in terms of the softmax function and the deep-learning mapping $\mathcal{N}_{\mathbf{x}}(\Theta)$, $\mathbf{p} := \frac{e^{\mathcal{N}_{\mathbf{x}}(\Theta)}}{\sum_{c=1}^{C} e^{(\mathcal{N}_{\mathbf{x}}(\Theta))_c}}$, where $e^{\mathcal{N}_{\mathbf{x}}(\Theta)}$ denotes a vector whose entries equal the exponential of the entries of

Figure D.1: Validation accuracy achieved by targets estimated via temporal ensembling using the cross entropy loss and our proposed cost function. The model is a ResNet-34 trained on CIFAR-10 with 40% symmetric noise. The temporal ensembling momentum $\beta$ is set to 0.7. Without the regularization term, the targets eventually overfit the noisy labels.

$\mathcal{N}_\mathbf{x}(\Theta)$. Plugging this into Eq. (20) yields

$$\nabla \mathcal{R}(\Theta) = \sum_{i=1}^{n} \frac{1}{1 - \langle \mathbf{p}, \mathbf{t} \rangle} \nabla \left( 1 - \frac{\langle e^{\mathcal{N}_\mathbf{x}(\Theta)}, \mathbf{t} \rangle}{\sum_{c=1}^{C} e^{(\mathcal{N}_\mathbf{x}(\Theta))_c}} \right) \tag{21}$$

$$= \sum_{i=1}^{n} \frac{-1}{1 - \langle \mathbf{p}, \mathbf{t} \rangle} \frac{\nabla \langle e^{\mathcal{N}_\mathbf{x}(\Theta)}, \mathbf{t} \rangle \cdot \sum_{c=1}^{C} e^{(\mathcal{N}_\mathbf{x}(\Theta))_c} - \langle e^{\mathcal{N}_\mathbf{x}(\Theta)}, \mathbf{t} \rangle \cdot \nabla \sum_{c=1}^{C} e^{(\mathcal{N}_\mathbf{x}(\Theta))_c}}{\left( \sum_{c=1}^{C} e^{(\mathcal{N}_\mathbf{x}(\Theta))_c} \right)^2} \tag{22}$$

$$= \sum_{i=1}^{n} \frac{-\nabla \mathcal{N}_\mathbf{x}(\Theta)}{1 - \langle \mathbf{p}, \mathbf{t} \rangle} \frac{e^{\mathcal{N}_\mathbf{x}(\Theta)} \odot \mathbf{t} \cdot \sum_{c=1}^{C} e^{(\mathcal{N}_\mathbf{x}(\Theta))_c} - \langle e^{\mathcal{N}_\mathbf{x}(\Theta)}, \mathbf{t} \rangle \cdot e^{\mathcal{N}_\mathbf{x}(\Theta)}}{\left( \sum_{c=1}^{C} e^{(\mathcal{N}_\mathbf{x}(\Theta))_c} \right)^2} \tag{23}$$

$$= \sum_{i=1}^{n} \frac{-\nabla \mathcal{N}_\mathbf{x}(\Theta)}{1 - \langle \mathbf{p}, \mathbf{t} \rangle} \left( \frac{e^{\mathcal{N}_\mathbf{x}(\Theta)} \odot \mathbf{t}}{\sum_{c=1}^{C} e^{(\mathcal{N}_\mathbf{x}(\Theta))_c}} - \frac{\langle e^{\mathcal{N}_\mathbf{x}(\Theta)}, \mathbf{t} \rangle}{\sum_{c=1}^{C} e^{(\mathcal{N}_\mathbf{x}(\Theta))_c}} \cdot \frac{e^{\mathcal{N}_\mathbf{x}(\Theta)}}{\sum_{c=1}^{C} e^{(\mathcal{N}_\mathbf{x}(\Theta))_c}} \right). \tag{24}$$

The formula can be simplified to

$$\nabla \mathcal{R}(\Theta) = \frac{-\nabla \mathcal{N}_\mathbf{x}(\Theta)}{1 - \langle \mathbf{p}, \mathbf{t} \rangle} \left( \mathbf{p} \odot \mathbf{t} - \langle \mathbf{p}, \mathbf{t} \rangle \cdot \mathbf{p} \right) \tag{25}$$

$$= \frac{\nabla \mathcal{N}_\mathbf{x}(\Theta)}{1 - \langle \mathbf{p}, \mathbf{t} \rangle} \begin{bmatrix} \mathbf{p}_1 \cdot (\langle \mathbf{p}, \mathbf{t} \rangle - \mathbf{t}_1) \\ \vdots \\ \mathbf{p}_C \cdot (\langle \mathbf{p}, \mathbf{t} \rangle - \mathbf{t}_C) \end{bmatrix} \tag{26}$$

$$= \frac{\nabla \mathcal{N}_\mathbf{x}(\Theta)}{1 - \langle \mathbf{p}, \mathbf{t} \rangle} \begin{bmatrix} \mathbf{p}_1 \cdot \sum_{k=1}^{C} (\mathbf{t}_k - \mathbf{t}_1) \mathbf{p}_k \\ \vdots \\ \mathbf{p}_C \cdot \sum_{k=1}^{C} (\mathbf{t}_k - \mathbf{t}_C) \mathbf{p}_k \end{bmatrix}. \tag{27}$$

# F  Algorithms

Algorithm 1 and Algorithm 2 provide detailed pseudocode for ELR combined with temporal ensembling (denoted simply by ELR) and ELR combined with temporal ensembling, weight averaging, two networks, and mixup data augmentation (denoted by ELR+) respectively. For CIFAR-10 and CIFAR-100, we use the sigmoid shaped function $e^{-5(1-i/40000)^2}$ ($i$ is current training step, following [41]) to ramp-up the weight averaging momentum $\gamma$ to the value we set as a hyper-parameter. For the other datasets, we fixed $\gamma$. For CIFAR-100, we also use previously mentioned sigmoid shaped function to ramp up the coefficient $\lambda$ to the value we set as a hyper-parameter. Moreover, each entry of the labels $y$ will also be updated by the targets $t$ using $\frac{y_c t_c}{\sum_{c=1}^{C} y_c t_c}$ in CIFAR-100.

**Algorithm 1:** Pseudocode for ELR with temporal ensembling.

---

**Require:** $\{\mathbf{x}^{[i]}, \mathbf{y}^{[i]}\}$, $1 \le i \le n$ = training data (with noisy labels)
**Require:** $\beta$ = temporal ensembling momentum, $0 \le \beta < 1$
**Require:** $\lambda$ = regularization parameter
**Require:** $\mathcal{N}_{\mathbf{x}}(\Theta)$ = neural network with trainable parameters $\Theta$

$\quad \mathbf{t} \leftarrow \mathbf{0}_{[n \times C]}$ $\qquad\qquad\qquad\qquad\qquad$ ▷ initialize ensemble predictions
$\quad$ **for** $t$ in $[1, num\_epochs]$ **do**
$\quad\quad$ **for** each minibatch $B$ **do**
$\quad\quad\quad$ **for** $i$ in $B$ **do**
$\quad\quad\quad\quad \mathbf{p}^{[i]} \leftarrow \mathcal{S}\left(\mathcal{N}_{\mathbf{x}_i}(\Theta)\right)$ $\qquad\qquad$ ▷ evaluate network outputs
$\quad\quad\quad\quad \mathbf{t}^{[i]} \leftarrow \beta \mathbf{t}^{[i]} + (1 - \beta)\mathbf{p}^{[i]}$ $\qquad$ ▷ temporal ensembling
$\quad\quad\quad$ **end for**
$\quad\quad\quad$ loss $\leftarrow -\frac{1}{|B|}\sum_{i=1}^{|B|}\sum_{c=1}^{C}\mathbf{y}_c^{[i]}\log\mathcal{S}\left(\mathcal{N}_{\mathbf{x}_i}(\Theta)\right)_c$ $\quad$ ▷ cross entropy loss component
$\quad\quad\quad\quad + \frac{\lambda}{|B|}\sum_{i \in B}\log\left(1 - \langle \mathcal{S}\left(\mathcal{N}_{\mathbf{x}_i}(\Theta)\right), \mathbf{t}^{[i]}\rangle\right)$ $\quad$ ▷ proposed regularization component
$\quad\quad\quad$ update $\Theta$ using stochastic gradient descent $\qquad$ ▷ update network parameters
$\quad\quad$ **end for**
$\quad$ **end for**
$\quad$ **return** $\Theta$

---

**Algorithm 2:** Pseudocode for ELR+.

---

**Require:** $\{\mathbf{x}^{[i]}, \mathbf{y}^{[i]}\}$, $1 \le i \le n$ = training data (with noisy labels)
**Require:** $\beta$ = temporal ensembling momentum, $0 \le \beta < 1$
**Require:** $\gamma$ = weight averaging momentum, $0 \le \gamma < 1$
**Require:** $\lambda$ = regularization parameter
**Require:** $\alpha$ = mixup hyperparameter
**Require:** $\mathcal{N}_{\mathbf{x}}(\Theta_1)$ = neural network 1 with trainable parameters $\Theta_1$
**Require:** $\mathcal{N}_{\mathbf{x}}(\Theta_2)$ = neural network 2 with trainable parameters $\Theta_2$

$\quad \mathbf{t}_1, \mathbf{t}_2 \leftarrow \mathbf{0}_{[n \times C]}, \mathbf{0}_{[n \times C]}$ $\qquad\qquad\qquad$ ▷ initialize averaged predictions
$\quad \bar{\Theta}_1, \bar{\Theta}_2 \leftarrow \mathbf{0}, \mathbf{0}$ $\qquad\qquad\qquad\qquad$ ▷ initialize averaged weights (untrainable)
$\quad$ **for** $t$ in $[1, num\_epochs]$ **do**
$\quad\quad$ **for** $k$ in $[1, 2]$ **do** $\qquad\qquad\qquad\qquad$ ▷ for each network
$\quad\quad\quad$ **for** each minibatch $B$ **do**
$\quad\quad\quad\quad \tilde{B} \leftarrow \text{mixup}(B, \alpha)$ $\qquad\qquad\qquad$ ▷ *mixup* augmentation on the mini-batch
$\quad\quad\quad\quad \bar{\Theta}_k = \gamma\bar{\Theta}_k + (1 - \gamma)\Theta_k$ $\qquad\quad$ ▷ weight averaging
$\quad\quad\quad\quad$ **for** $i$ in $B$ **do**
$\quad\quad\quad\quad\quad \mathbf{p}^{[i]} \leftarrow \mathcal{S}\left(\mathcal{N}_{\mathbf{x}_i}(\bar{\Theta}_{\{1,2\}\backslash k})\right)$ $\qquad$ ▷ network evaluation with weight averaging
$\quad\quad\quad\quad\quad \mathbf{t}_k^{[i]} \leftarrow \beta\mathbf{t}_k^{[i]} + (1 - \beta)\mathbf{p}^{[i]}$ $\qquad$ ▷ temporal ensembling
$\quad\quad\quad\quad$ **end for**
$\quad\quad\quad\quad$ loss $\leftarrow -\frac{1}{|B|}\sum_{i=1}^{|B|}\sum_{c=1}^{C}\mathbf{y}_c^{[i]}\log\mathcal{S}\left(\mathcal{N}_{\tilde{\mathbf{x}}_i}(\Theta_k)\right)_c$ $\quad$ ▷ cross entropy loss component
$\quad\quad\quad\quad\quad + \frac{\lambda}{|B|}\sum_{i \in B}\log\left(1 - \langle \mathcal{S}\left(\mathcal{N}_{\tilde{\mathbf{x}}_i}(\Theta_k)\right), \tilde{\mathbf{t}}^{[i]}\rangle\right)$ $\quad$ ▷ proposed regularization component
$\quad\quad\quad\quad$ update $\Theta_k$ using SGD $\qquad\qquad\qquad$ ▷ update network parameters
$\quad\quad\quad$ **end for**
$\quad\quad$ **end for**
$\quad$ **end for**
$\quad$ **return** $\Theta_1, \Theta_2$

---

To apply *mixup* data augmentation, when processing the $i$th example in a mini-batch $(\mathbf{x}^{[i]}, \mathbf{y}^{[i]}, \mathbf{t}^{[i]})$, we randomly sample another example $(\mathbf{x}^{[j]}, \mathbf{y}^{[j]}, \mathbf{t}^{[j]})$, and compute the $i$th mixed data $(\tilde{\mathbf{x}}^{[i]}, \tilde{\mathbf{y}}^{[i]}, \tilde{\mathbf{t}}^{[i]})$ as follows:

$$\ell \sim \text{Beta}(\alpha, \alpha),$$
$$\ell' = \max(\ell, 1 - \ell),$$
$$\tilde{\mathbf{x}}^{[i]} = \ell'\mathbf{x}^{[i]} + (1 - \ell')\mathbf{x}^{[j]},$$
$$\tilde{\mathbf{y}}^{[i]} = \ell'\mathbf{y}^{[i]} + (1 - \ell')\mathbf{y}^{[j]},$$
$$\tilde{\mathbf{t}}^{[i]} = \ell'\mathbf{t}^{[i]} + (1 - \ell')\mathbf{t}^{[j]},$$

| Data set | Train | Val | Test | Image size | # classes |
|---|---|---|---|---|---|
| Datasets with Clean Annotation | | | | | |
| CIFAR-10 | 45K | 5k | 10K | $32 \times 32$ | 10 |
| CIFAR-100 | 45K | 5k | 10K | $32 \times 32$ | 100 |
| Datasets with Real World Noisy Annotation | | | | | |
| Clothing-1M | 1M | 14K | 10K | $224 \times 224$ | 14 |
| Webvision1.0 | 66K | - | 2.5K | $256 \times 256$ | 50 |

Table G.1: Description of the datasets used in our computational experiments, including the training, validation and test splits.

where $\alpha$ is a fixed hyperparameter used to choose the symmetric beta distribution from which we sample the ratio of the convex combination between data points.

# G   Description of the Computational Experiments

Source code for the experiments is available at https://github.com/shengliu66/ELR.

## G.1   Dataset Information

In our experiments we apply ELR and ELR+ to perform image classification on four benchmark datasets: CIFAR-10, CIFAR-100, Clothing-1M, and a subset of WebVision. Because CIFAR-10, CIFAR-100 do not have predefined validation sets, we retain 10% of the training sets to perform validation. Table G.1 provides a detailed description of each dataset.

## G.2   Data preprocessing

We apply normalization and simple data augmentation techniques (random crop and horizontal flip) on the training sets of all datasets. The size of the random crop is set to be consistent with previous works [56, 17]: 32 for the CIFAR datasets, $224 \times 224$ for Clothing1M (after resizing to $256 \times 256$), and $227 \times 227$ for WebVision.

## G.3   Training Procedure

Below we describe the training procedure for ELR (i.e. the proposed approach with temporal ensembling) for the different datasets. The information for ELR+ is shown in Table G.4. In ELR+ we ensemble the outputs of two networks during inference, as is customary for methods that train two networks simultaneously [22, 14].

**CIFAR-10/CIFAR-100**: We use a ResNet-34 [15] and train it using SGD with a momentum of 0.9, a weight decay of 0.001, and a batch size of 128. The network is trained for 120 epochs for CIFAR-10 and 150 epochs for CIFAR-100. We set the initial learning rate as 0.02, and reduce it by a factor of 100 after 40 and 80 epochs for CIFAR-10 and after 80 and 120 epochs for CIFAR-100. We also experiment with cosine annealing learning rate [26] where the maximum number of epoch for each period is set to 10, the maximum and minimum learning rate is set to 0.02 and 0.001 respectively, total epoch is set to 150.

**Clothing-1M**: We use a ResNet-50 pretrained on ImageNet same as Refs. [45, 47]. The model is trained with batch size 64 and initial learning rate 0.001, which is reduced by $1/100$ after 5 epochs (10 epochs in total). The optimization is done using SGD with a momentum 0.9, and weight decay 0.001. For each epoch, we sample 2000 mini-batches from the training data ensuring that the classes of the noisy labels are balanced.

**WebVision**: Following Refs. [17, 22], we use an InceptionResNetV2 as the backbone architecture. All other optimization details are the same as for CIFAR-10, except for the weight decay (0.0005) and the batch size (32).

## G.4   Hyperparameters selection

We perform hyperparameter tuning on the CIFAR datasets via grid search: the temporal ensembling parameter $\beta$ is chosen from $\{0.5, 0.7, 0.9, 0.99\}$ and the regularization coefficient $\lambda$ is chosen from $\{1, 3, 5, 7, 10\}$ using the validation set. The selected values are $\beta = 0.7$ and $\lambda = 3$ for symmetric noise, $\beta = 0.9$ and $\lambda = 1$ for assymetric noise on CIFAR-10, and $\beta = 0.9$ and $\lambda = 7$ CIFAR-100. For Clothing1M and WebVision we use the same values as for CIFAR-10. As shown in Section H, the performance of the proposed method seems to be robust to changes in the hyperparameters. For ELR+, we use the same values for $\lambda$ and $\beta$. The *mixup* $\alpha$ is set to

|  | CIFAR-10 | CIFAR-100 | Clothing-1M | Webvision |
|---|---|---|---|---|
| batch size | 128 | 128 | 64 | 32 |
| architecture | PreActResNet-18 | PreActResNet-18 | ResNet-50 (pretrained) | InceptionResNetV2 |
| training epochs | 200 | 250 | 15 | 100 |
| learning rate (lr) | 0.02 | 0.02 | 0.002 | 0.02 |
| lr scheduler | divide 10 at 150th epoch | divide 10 at 200th epoch | divide 10 at 7th epoch | divide 10 at 50th epoch |
| weight decay | 5e-4 | 5e-4 | 1e-3 | 5e-4 |

Table G.2: Training hyperparameters for ELR+ on CIFAR-10, Clothing-1M and Webvision.

Figure H.1: Test accuracy on CIFAR-10 with symmetric noise level 60%. The mean accuracy over four runs is reported, along with bars representing one standard deviation from the mean. In each experiment, the rest of hyperparameters are fixed to the values reported in Section G.4.

1 (chosen from $\{0.1, 2, 5\}$ via grid search on the validation set) and the value of the weight averaging parameter $\gamma$ is set to $0.997$ (which is the default value in the public code of Ref. [41]) except Clothing1M, which is set to $0.9999$.

# H   Sensitivity to Hyperparameters

The main hyperparameters of ELR are the temporal ensembling parameter $\beta$ and regularization coefficient $\lambda$. As shown in the left image of Figure H.1, performance is robust to the value of $\beta$, although it is worth noting that this is only as long as the momentum of the moving average is large. The performance degrades to 38% when the model outputs are used to estimate the target without averaging (i.e. $\beta = 0$). The regularization parameter $\lambda$ needs to be large enough to neutralize the gradients of the falsely labeled examples but also cannot be too large, to avoid neglecting the cross entropy term in the loss. As shown in the center image of Figure H.1, the sensitivity to $\lambda$ is also quite mild. Finally, the right image of Figure H.1 shows results for ELR combined with mixup data augmentation for different values of the mixup parameter $\alpha$. Performance is again quite robust, unless the parameter becomes very large, resulting in a peaked distribution that produces too much mixing.

# I   Training Time Analysis

In Table I.1 we compare the training times of ELR and ELR+ with two state-of-the-art methods, using a single Nvidia v100 GPU. ELR+ is twice as slow as ELR. DivideMix takes more than 2 times longer than ELR+ to train. Co-teaching+ is about twice as slow as ELR+.

| Co-teaching+[52] | DivideMix[22] | ELR | ELR+ |
|---|---|---|---|
| 4.4h | 5.4h | 1.1h | 2.3h |

Table I.1: Comparison of total training time in hours on CIFAR-10 with 40% symmetric label noise.