[Reviews · NeurIPS 2020]

Review 1

Summary and Contributions: This paper studies the memorization phenomenon for learning with noisy labels. They theoretically analyze when and why memorization happens, and proposes a method based on semi-supervised learning to address it. Experiments are performed on multiple benchmarks. ** Final review** The authors address my questions well in their response. After reading other reviewers' comments, I'll keep my original score and lean towards accept.

Strengths: 1. This paper is well motivated. It's nice to see in Figure 1 that memorization is prevented. 2. The theoretical analysis sheds some lights on previous methods which use model's output to modify the target. 3. The paper is mostly well-written and easy to understand.

Weaknesses: 1. The memorization effect is not new to the community. Therefore, the novelty of this paper is not sufficiently demonstrated. The authors need to be clearer what extra insights this paper gives. 2. It would be better if the author could provide some theocratical justification in terms of why co-training and weight averaging can improve results, since they are important for the performance. 3. The empirical performance does not seem to be very strong compared to DivideMix. Some explanations are needed.

Correctness: Yes the claims and method are correct.

Clarity: Yes it is well-written in general.

Relation to Prior Work: Yes it is clearly discussed.

Reproducibility: Yes

Additional Feedback:


Review 2

Summary and Contributions: One folklore result in deep learning today is that deep neural networks (DNNs) learn the correct labels first before they memorize the examples with noisy labels. One contribution of this paper is to argue that this phenomenon is not restricted to DNNs, by arguing theoretically that a similar behavior can occur in logistic regression in the over-paramterized regime. In turn, this result inspired the development of a new algorithm for learning with noisy labels. The authors show experimentally that their method is competitive with state-of-the-art methods. ============= Post-rebuttal comments: The score was updated based on the authors' response. I had a major concern regarding the proof but the authors' have clarified it in their rebuttal. However, my statement regarding the size of the spheres remains the same but it is not a major objection.

Strengths: Understanding the behavior of deep neural network is a very important research topic. The authors aim to shed some light into one particular empirical observation, namely that examples with correct labels are learned first, by arguing that it holds for simpler models as well, such as logistic regression. Inspired by this, they propose a novel algorithm for learning with noisy labels. Informally, the papers points out that the aforementioned empirical observation has a simple explanation. During the early stage of learning, correct labels will steer gradient descent towards the correct direction whereas noisy labels tend to have a smaller magnitude (since they tend to cancel each other and they are fewer in number). During the later stage of learning, on the hand, the examples with the correct labels will have a small gradient, since they are classified correctly, so gradient descent will be steered mainly by the noisy examples. This is why correct labels are learned first while noisy labels are memorized last. To fix this issue, the author propose adding a regularization term to the objective function. Given the early epochs of training, one may construct an approximation of the true labels upon knowing that noisy labels are learned last, e.g. using temporal averaging. With this "new" target t, a regularization term is added so that the prediction of the model does not deviate much from t during the later stages of learning (see Eq 9). Of course, in order for this to work, one has to estimate the true probabilities accurately, and the authors show that previous methods, such as mixup and co-teaching, can help significantly.

Weaknesses: I have many reservation against the claims of the paper. I would appreciate it if the authors can clarify some of these issues during their rebuttal. First, the proof of their main theorem about logistic regression has many issues. One key issue is that the authors make assumptions within the proof that are not clearly stated or justified upfront. For example, in Line 440 in the supplementary materials, the proof assumes that theta^Tv<.1. Where did this assumption come from? There is no reason to expect it to hold apriori. It seems to have been added just to close a hole in the proof but that means Proposition 3 was not really "proven". What is worse is that later in the proof in Line 457, the authors make the *opposite* assumption! A second issue with the theorem is that it assumes a highly contrived setting for logistic regression. First, the claim is that it holds for a "sufficiently small sigma." Basically, the authors show that if we have two spheres (one for the positive class and one for the negative class) and both spheres are *sufficiently tiny* (i.e. all instances are almost identical), then correct labels will be learned first while noisy labels will be learned last. I think this is not surprising since all instances are almost identical and correct labels correspond to the majority in a very tiny ball. The authors argue that they can extend their result to the case where Delta>1/2 but I don't think that is possible. If Delta?1.2, then the role of v and -v would be interchanged (since the majority of labels are flipped). Also, the authors assume that the number of examples n is very close to the problem dimension d. In particular, they assume that 0.75n<d<n. Second, the empirical support the authors provide in Figures A1 and B1 does not actually show the main phenomenon that was cited earlier for deep learning. The main interesting phenomenon is that noisy labels tend to be classified correctly early during training before their noisy labels are memorized. So, the accuracy within the noisy set is not a monotone function: it goes up initially before it drops. This is shown in Figure 1:Top-right. Figures A1 and B1 for regression do not show that phenomenon. Memorization in Figure A.1 takes effect almost immediately after very few iterations. Finally, the algorithm does not outperform state-of-the-art methods. Given the concerns above and the fact that there is no significant improvement experimentally, I think the paper is not suitable for publication at NeurIPS.

Correctness: No, I think there are issues with the proofs and the setting. Please see my detailed comments above.

Clarity: Yes, the paper is well-written.

Relation to Prior Work: Yes, relation to previous works is clearly discussed.

Reproducibility: Yes

Additional Feedback: 1- Some of the figures contain redundant information, which can be confusing. For example, the two curves in Figure 1 left convey the same information (blue = 1 - green). Having one curve would be better. The same holds for Figure A1. Also, in Figure B1, the right column can be inferred from the left column (right column is the absolute value of the left). 2- Putting the issues with the proof of the theorem aside, the method proposed here is actually independent of the argument for logistic regression. It is inspired by the form of the gradient in Eq 5, the heuristic argument in Section 4.1 , and the empirical support in Figure 2. I don't think including the result on logistic regression adds value and the paper would be stronger without it. 3- - The experiments in Table 1 show an improvement of ELR but it is not clear if the improvement is due to the algorithm itself (i.e. the regularization term) or the improvement is due to the new target t, which is estimated independently of ELR. It is possible that ELR is not really needed. There are two typos: - In Line 46, I think the authors meant to say that they do NOT assume that the correct classes are known. - In Line 109, the minimization of Theta is over the space R^(2xp), not R^p.


Review 3

Summary and Contributions: This paper focuses on coping with label noise by exploiting a regularization term. This term seeks to maximize the inner product between the output of model and the targets, which prevents memorization of noisy labels. The authors provide a detailed theoretical analysis for memorization effect of neural networks. Extensive experiments are conducted to verify the effectiveness of the proposed method. **After reading author response**: I have read the response and comments from other reviewers. The author cleared my doubts well. Thus, I will keep my initial score.

Strengths: 1. The writing logic of this paper is good. 2. Complete and meaningful theoretical proof. 3. Sufficient experimental results which validate the effectiveness of ELR or ELR+, especially in noisy synthetic datasets.

Weaknesses: 1. Some experimental results are not convincing. In the experiments on real-world dataset, the advantage of the proposed method is weak. On Clothing1M, ELR and ELR+ employ some tricks to get better performance. On ILSVRC12, DivideMix performs much better than ELR and ELR+. 2. There are many hyper-parameters needed to be considered. As shown in Figure G.1, the proposed method is sensitive to \lambda. 3. Some details should be added. For example, when ELR or ELR+ achieves worse performance than baselines, the authors should discuss the potential problem of the proposed method or analyze the reason for this phenomenon.

Correctness: Yes. The claims and method are correct. The empirical methodology is correct.

Clarity: Yes, the paper is well written.

Relation to Prior Work: Yes. It is clearly discussed.

Reproducibility: Yes

Additional Feedback:


Review 4

Summary and Contributions: This work introduces a novel regularization method to prevent memorization of false labels in classification tasks. The authors show that early learning followed by memorization of false labels occurs in simple linear generative models then argue that deep non-linear models show similar behavior. The proposed regularization method relies on techniques from semi-supervised learning to reduce the gradients of noisy examples by using a running average of the model's outputs. This preserves the correct classifications of noisy labels that the authors claim are present in the early learning phase. The authors propose additional extensions: computing the running average of model outputs from an ensemble of past models using weight averaging, predicting targets from outputs of two separate networks, and the use of mix-up data augmentation. The authors demonstrate the technique is effective and produces competitive results on various synthetic and real datasets. **Upon reviewing author responses**: I still believe the paper should be accepted, so I will keep my score at 7.

Strengths: * The authors present a simple linear model example that illustrates the concept of the early learning phase and helps the reader understand how this can lead to memorization when some fraction of the dataset contains noisy labels. This analysis is a nice stepping stone for the reader to the regularization technique they propose. * The authors motivate the definition of the regularization term with a toy example and plots that illustrate the effect of this regularization and how it balances with the cross entropy term. * The authors show promising results with ResNet-34 on CIFAR-10 and CIFAR-100 with symmetric and asymmetric label noise, setting a new state of the art benchmark on this task amongst techniques that only modify the training loss. * The proposed method is competitive with techniques that modify sample selection and data augmentation on CIFAR-10. The SOTA technique DivideMix that they compare with is significantly more complicated than the proposed regularization technique, requiring at least double the computational complexity of a given model due to the need to train multiple versions of the model. I am unfamiliar with this area so I cannot speak to the novelty of the technique and its relation to prior work. Improving classifiers and principled approaches to handling the real-world problem of label noise is of great interest to ML practitioners in the NeurIPS community. Improving understanding of training dynamics in classification models is of interest to theorists.

Weaknesses: * Implementation requires a parameter (the target vector) that scales with the size of the dataset. This may be impractical for very large datasets.

Correctness: I did not notice any correctness issues with the claims and method; however, I am inexperienced in this area.

Clarity: The paper is easy to understand and progressively grows the reader's understanding with a mix of plain descriptions in the prose and light derivations with pointers to the appendix where relevant. I appreciated the writing style.

Relation to Prior Work: The related work section appears robust, but I am not experienced in this domain.

Reproducibility: Yes

Additional Feedback: Code samples would aid in reproducibility, but the method seems as though it would be easy to implement from the authors' description.

[Author Response · NeurIPS 2020]

We are grateful to the reviewers for their time and their thoughtful comments, which we believe will improve the paper. We first clarify the comparison with DivideMix and then address all individual comments below. DivideMix is an impressively engineered combination of **multiple different techniques** with associated hyperparameters: a warm-up period, a mixture model, two networks trained with multiple augmentations, a sharpening function, and a confidence penalty. In contrast we propose a **novel regularization term**, grounded in our theoretical analysis, which achieves strong performance **on its own**. Due to its complexity DivideMix takes 2 to 4 times longer to train than our methods (e.g. 1.1h for ELR, 2.3h for ELR+ vs 5.4h for DivideMix for CIFAR-10 on a single Nvidia v100 GPU). Moreover, the results reported for DivideMix on CIFAR10 are obtained by monitoring accuracy on the validation set during training and choosing the highest value, i.e. **not on completely held-out data**. In contrast, we use 10% of the training set for validation, and treat the validation set as a **purely held-out test set** (this also means that we train on less data). Following the same approach as DivideMix, the accuracy for ELR+ is the same or even higher (e.g. 94.6%, 93.5%, 76.5% for 50%, 80%, 90% noise levels vs 94.6%, 93.2%, 76% for DivideMix) but we were not comfortable reporting best validation accuracy, which may overfit the validation set. If we further combine ELR+ with the *unsupervised loss component* in Ref.[20] we outperform DivideMix clearly at high noise levels (94.2%, 79.8% for 80%, 90% respectively), achieving **state-of-the-art performance**. However our focus is to propose a **new tool**, not to combine many different ones to maximize performance. Despite this, ELR+ slightly outperforms DivideMix on Clothing1M and WebVision. We believe that the superior performance of DivideMix on ILSVRC12 is due to the use of semi-supervised learning techniques. We will explain all of this in our revision, including possible limitations as suggested by Reviewer 3.

**Reviewer 1** • *The memorization effect is not new to the community.* - We agree. Our contribution is to show that this effect is actually a **fundamental phenomenon in high-dimensional classification** that does not only exist in nonlinear models (e.g. neural networks) and provide a **rigorous mathematical characterization** on linear models.
• *... theoretical justification in terms of why co-training and weight averaging can improve results* - This is a great suggestion. We believe that it may be due to a reduction in confirmation bias. We will explain this in the paper.

**Reviewer 2** We thank the reviewer for the very careful examination of the proofs for Section 2, and for drawing our attention to several points that should be explained more clearly.
• *In Line 440..., the proof assumes that $\theta_t^\top v < .1$... In Line 457, the authors make the opposite assumption [$\theta_T^\top \mathbf{v} \geq .1$]* - This is a crucial point: $T$ is defined to be the first time at which $\theta_T^\top \mathbf{v} \geq .1$, i.e. the **end of the early-learning stage** during which the weights become aligned with the true direction (Prop.3). At that point, enough correlation is achieved to produce accurate predictions (Theorem 4). However, at the same time the gradients from the true labels decrease (Prop.5), which eventually leads to memorization (Theorem 6). We will explain this more clearly.
• *both spheres are sufficiently tiny (i.e. all instances are almost identical)* - Surprisingly and counter-intuitively the spheres are **not tiny** due to high-dimensional geometry. The clusters are, approximately, two spheres whose centers are 2 units apart and whose radii are $\sigma\sqrt{p}$, where $p$ is the dimension. When $p$ is large, $\sigma\sqrt{p} \gg 2$, so the spheres are huge compared to the distance between the two classes. We will clarify this important point in our revision.
• *If $\Delta > 1/2$...the majority of labels are flipped* - As defined in our paper, with probability $\Delta$, each label is replaced by a **random** label, not the wrong label. Therefore a $1 - (\Delta/2)$ proportion of the examples are correctly labeled, and the majority of labels are correct for all $\Delta < 1$. We will make this clear.
• *n is very close to the problem dimension [p]* - In high-dimensional statistics it is common to focus on the case that $n$ and $p$ are of the same order. We can always assume that $p \leq n$ by restricting our attention to the $n$-dimensional span of the training data. If $p \ll n$, then we are in the "low-dimensional" regime, where many of the surprising features of modern ML (such as the early-learning and memorization phenomena we investigate) are known not to occur.
• *the accuracy within the noisy set ... goes up initially before it drops ... not observed in Figure A1 and B1 for linear models* - A close look at the first few iterations of the top right graph in Fig. A1 and B1 shows that there is a decreasing trend of the accuracy for the linear model as well, it is just faster than in the nonlinear model (probably because the linear model does not need to learn complex features). We will adjust the scale of x axis so that this can be seen better.
• *Some of the figures contain redundant information* - We agree and will update the figures.
• *I don't think including the result on logistic regression adds value* - The method is inspired by the observation that the gradient of the deep-learning loss decouples into two parts, one of which is identical to the one in logistic regression. We believe that a rigorous analysis is therefore valuable, and may be helpful to some readers (e.g. Reviewer 4).
• *it is not clear if the improvement is due to the regularization term or ... due to the new target* - This is an excellent point! We will add a graph showing that the accuracy of the targets without regularization is significantly lower (e.g. 77.43% vs 86.12% for CIFAR10 60% noise).

**Reviewer 3** • *There are too many hyper-parameters to consider ... sensitive to $\lambda$* - We have 2 parameters for ELR and 4 parameters for ELR+ (DivideMix has 6), and they do not need to be heavily tuned. As mentioned in Section F.3, our results on CIFAR-10, Clothing-1M, and Webvision are obtained with the **exact same** hyper-parameter values (it is possible that tuning them would further increase performance). It is natural for there to be a dependence on $\lambda$, but $0.3 \leq \lambda \leq 0.5$ achieves essentially the same performance on CIFAR-10, as shown in Fig. G.1.

**Reviewer 4** • *... target vector scales with the size of the dataset* - This is correct, but one can store the target vector on disk (along with the labels) and load it whenever required.

[Meta-Review · NeurIPS 2020]

The paper studies the following interesting phenomenon (observed in the previous literature): when trained on the dataset with incorrectly labeled points (i.e. "label noise"), DNNs first learn the benign ("correctly labeled") points and once this is done they start "memorizing" the noisy points. It was previously shown in the literature (empirically) that the second "memorization" phase hurts the generalization. The authors make 2 Contributions: (Contribution 1) They demonstrate (empirically and theoretically) that similar phenomenon can be observed in the simpler setting of the over-parametrized (dimensionality ~ number of points) linear two-class logistic regression, when the class distributions are isotropic Gaussian with fixed means $\pm mu$ and vanishing variance (see Theorem 1 and Figure A.1). (Contribution 2) Motivated by the theory of contribution 1, the authors propose a novel regularizer. (see Eq. 6). When used in the vanilla DNN training with the cross-entropy loss, this regularizer successfully prevents the networks from falling to the "memorization phase" (as evidenced by Figure 1). All the reviewers agree that the topic and the focus of this paper is very timely. The questions related to DNN and "memorization" has surfaced recently in many works on DNNs, both theoretical and applied. Apart from providing theoretical insights on the empirical phenomenon (Contribution 1), this paper is one of the first works (to be best of our knowledge) that propose to utilize it explicitly during DNN training. The proposed method is a very reasonable first attempt in this direction ("DNN makes good decisions in the beginning of the training, before it starts memorizing, so let's influence its decisions in the later stages by the ones it made in the earlier stages"). While the proposed strategy has many moving parts (when exactly DNN switches from the first to the second phase? How exactly should we bias later stages? i.e. how should we choose "targets" in Eq. 6), the authors presented several particular design choices that empirically lead to strong results. Even though the proposed method does not outperform SOTA results across the table, it is competitive while requiring much fewer tricks than other methods. Finally, even though the theory (Theorem 1) holds only for the particular case of the linear two-class logistic regression, it motivates the choice of the regularizer that (in turn) performs well in the realistic practical settings. This shows that the paper contains insights that will be likely useful for the future research in this direction. However, the reviewers had several concerns. The following three are among the most important ones to address: (1) Rev#2 points out that Theorem 1 essentially assumes that the classes are supported on tiny compact subsets very well separated from each other. In the rebuttal, the authors claim "when $p$ is large, $\sigma \sqrt{p} >> 2$, so the spheres are huge". Some of the reviewers (myself included) were not convinced with this argument. The text of the paper never makes explicit what the assumptions mean exactly when they say "variance is sufficiently small". It may be that there is a relation between dimensionality $p$ and the variance $\sigma$ implicitly assumed in the proofs that forbid $\sigma \sqrt{p}$ to become large (as in the argument from the rebuttal). (2) For instance, in the Eq. 11 from the supplementary we see that $\sigma \|\theta_0 - \theta_t\|$ is required to be close to zero, yet the authors don't prove formally that this is possible. This step should be made precise! One naive way of doing this that comes to my mind is to show that $\|\theta_0 - \theta_t\| ~ O(\sqrt{p})$ and set $\sigma ~ o(1/\sqrt{p})$. However, this would contradict $\sigma \sqrt{p} >> 2$ stated in the rebuttal. (3) More discussion on the hyperparameter selection for ELR and ELR+ are required. Concerns 1 and 2 sketched above are not critical, as they don't directly affect the proposed method (Contribution 2). However, they need to be carefully addressed in order to save Contribution 1. In particular, the authors are required to put *exact* assumptions on $\sigma$ explicitly in the text of Theorem 1. Most likely (if my guess from Concern 2 above is correct), the authors will have to write something like $\sigma \sqrt{p} ~ o(1)$, in which case Rev#2's original intuition will be correct and the result indeed holds only if the classes are far away from each other and consist of almost identical points. I think this fact won't diminish Contribution 1, after all (i) it still provides a simple setting where the phenomenon can be observed and (ii) it still provides a reasonable motivation for the proposed regularizer. I am tending towards acceptance, given the authors address all the concerns summarized in this meta-review (as well as other minor concerns mentioned by the reviewers).